# High-temperature probe of electron compressibility via asymmetric Coulomb drag

Yingjia Liu[1,2,3], Kaining Yang[4,5,6], Hanwen Wang[3], Qin Zhang[3], Hongpeng Liu[3], Kenji Watanabe [7], Takashi Taniguchi [8], Wencai Ren [1,2] ✉, Zheng Vitto Han [3,4,5,6] ✉ & Siwen Zhao [3] ✉

Lateral charge transport of a two-dimensional (2D) electronic system can be much influenced by feeding a current into another closely spaced 2D conductor, known as the Coulomb drag phenomenon – a powerful probe of electron-electron interactions and collective excitations. Here, we show that Coulomb drag in a deliberately asymmetric van der Waals bilayer can serve as a layer-selective probe of electronic compressibility that remains invisible to standard transport. We devise a $MoS_2$/graphene double layer with large disparity in effective mass and Fermi temperature between them, separated by a ~ 3 nm hexagonal boron nitride spacer, and operate in the degenerate Fermi liquid regime. The $MoS_2$ drag channel exhibits constant electronic compressibility and acts as a sensitive transducer of graphene's Landau-level physics at finite magnetic fields. At elevated temperatures and moderate magnetic fields, clear Shubnikov-de Haas-like behaviour in the drag signal tracks the quantum oscillation in compressibility of graphene even when its own magnetotransport remains essentially featureless under the same conditions. Our results establish asymmetric Coulomb drag as a compressibility spectroscopy for 2D systems, enabling access to quantum phenomena that may leave only weak, or even negligible, fingerprints in transport.

In two closely spaced low dimensional conductors, charge carriers driving in one active layer is often observed to induce drag characteristics in another passive layer, yielding a current or voltage in the latter. Such effects offer a fundamental yet direct probe for electronic momentum and/or energy exchange via long range Coulomb interactions, as well as many-body physics beyond single-particle transport[1]. Indeed, Coulomb drag phenomena have been extensively manifested in different regimes, including quantum wells or graphene separated with large distance in the weak coupling limit[2,3], and excitonic condensation when interlayer charge carriers are matched in the quantum Hall limit[4–12]. More recently, emerging physical phenomena are also reported in exotic drag between graphene and superconductors, topological insulators, 1D-1D Luttinger liquid, quantum dots and mixed dimensional electrons[13–18].

[1]Shenyang National Laboratory for Materials Science, Institute of Metal Research, Chinese Academy of Sciences, Shenyang, China. [2]School of Material Science and Engineering, University of Science and Technology of China, Anhui, China. [3]Liaoning Academy of Materials, Shenyang, China. [4]State Key Laboratory of Quantum Optics Technologies and Devices, Institute of Optoelectronics, Shanxi University, Taiyuan, China. [5]Collaborative Innovation Center of Extreme Optics, Shanxi University, Taiyuan, China. [6]Institute of Carbon-Based Thin Film Electronics, Peking University, Shanxi (ICTFE-PKU), Taiyuan, China. [7]Research Center for Electronic and Optical Materials, National Institute for Materials Science, 1-1 Namiki, Tsukuba, Japan. [8]Research Center for Materials Nanoarchitectonics, National Institute for Materials Science, 1-1 Namiki, Tsukuba, Japan. ✉e-mail: wcren@imr.ac.cn; vitto.han@gmail.com; siwenzhao0126@gmail.com

Among those reported, gapped two-dimensional (2D) semiconductors, with inherently large correlations in the massive carriers, have been a missing piece in the jigsaw puzzle of various drag regimes. Especially, a peculiar family of massive-massless double layers has remained largely unvisited. Taking the Wigner-Seitz radius $r_s$ (strong correlation when $r_s > 10$) as a measure of interaction strength in 2D electron systems, massless Dirac fermions in monolayer graphene has a density-independent value of $r_s \sim 0.7 - 0.8$[19,20]. Meanwhile, in bilayer graphene and conventional 2D electron gases in quantum wells, Fermi surfaces are well defined and $r_s$ is sufficiently large only when carrier density is remained ultra low ($< 10^{10}$ cm$^{-2}$), which is manifested in such as an unconventional negative frictional drag in the vicinity of charge neutral in double graphene bilayers[21,22]. Gapped 2D semiconductors, the transition metal dichalcogenides (TMDs) for instance, host massive fermions and relatively constant magneto-responses in resistivity over a broad range of temperature[23,24]. Thus, the interplay of these asymmetric drag paradigm, i.e., massive interacting Schrödinger fermions with massless Dirac fermions, is expected to unveil new physical phenomena, yet its experimental access has been rare, so far[25–27]. This is mainly due to the grand challenge of obtaining Ohmic contacts and maintaining high-mobility charge transport at their low temperature ground states.

In this work, we demonstrate large Coulomb drag responses in a semiconductor-semimetal hybrid, realized in a MoS$_2$-graphene heterostructure separated by an ultrathin 3 nm h-BN dielectric. Using a 2D window contact method, Ohmic contacts are realized in MoS$_2$ throughout the temperature range tested in this study. Unlike conventional drag systems, we observe a drag resistance ($R_{drag}$) as high as several hundred Ω, with a passive-to-active drag ratio (PADR) reaching ~ 0.6, orders of magnitude larger than previously reported values[28,29]. Furthermore, we identify a well-defined Fermi-liquid like phase of drag responses by examing systematically the temperature dependence, as well as the carrier density dependence of $R_{drag}$. The MoS$_2$ drag layer has an essentially constant electronic compressibility and thereby serves as a high-sensitivity transducer of graphene's Landau quantization at finite magnetic field. At elevated temperatures (such as above liquid nitrogen temperature) and moderate magnetic fields, the drag signal exhibits clear Shubnikov-de Haas (SdH)-like oscillations that track graphene's compressibility, even when graphene's own transport fails to deliver any information at such temperature and magnetic fileds. These observations highlight the crucial role of interlayer correlations of drag in amplifying the electron compressibility in such an asymmetric (in terms of compressibility in MoS$_2$ and graphene) drag system. Our findings of drag-enhanced readout of graphene's Landau levels using a flat-compressibility MoS$_2$ transducer may offer insights into designing next-generation interaction-driven electronic devices.

## Results and discussion
### Fabrications and characterizations of MLG-MoS$_2$ drag devices
Monolayer graphene, bilayer MoS$_2$ and h-BN flakes were mechanically exfoliated from bulk crystals. As illustrated in Fig. 1a, the van der Waals heterostructure is stacked using the dry transfer method[30], and then encapsulated by top and bottom h-BN flakes, with the top h-BN etched into micron-metre sized 2D windows. A windowed contact method is thus employed to achieve Ohmic contacts to the MoS$_2$ channel throughout the temperature range from 0.3 K to 300 K[31]. This requires MoS$_2$ to be the top layer in the heterostructure to facilitate the fabrication process in this study. The devices were equipped with dual metallic gates and electrodes of Ti/Au via standard lithography and electron-beam evaporation (fabrication details are available in Methods). More detailed fabrication processes can be seen in Supplementary Figs. 1–3. We found that different bottom gate geometry will affect the Coulomb drag measurements (Supplementary Fig. 4), and the main text will focus on the geometric configuration as illustrated in Supplementary Fig. 1.

Figure 1a describes the essential nanostructure in this study: a semiconductor − semimetal drag hybrid, realized in a MoS$_2$-graphene double layer separated by an ultrathin 3 nm h-BN dielectric. Here, considering the low energy physics at the Fermi level within the solid state gate doping range, charge carriers in MoS$_2$ and graphene are massive Schrödinger and massless Dirac fermions, respectively. Figure 1b shows the optical micrograph of a typical drag device (sample-S21), with the corresponding fabrication flow shown in Supplementary Fig. 3. A bright-field scanning transmission electron microscopy (STEM) image and the corresponding electron energy loss spectroscopy (EELS) mapping in Fig. 1e − g clearly reveal the cross-sectional structure of a typical drag device with bilayer MoS$_2$ as the active (or passive) layer and monolayer graphene as the passive (or active) layer. Within the device, carriers in each layer can be tuned independently. For instance, at $T$ = 200 K, typical field effect curves in the MoS$_2$ channel (Fig. 1c) and in the graphene channel (Fig. 1d) can be obtained, respectively. The $T$-dependent transfer curves of graphene and MoS$_2$ in Supplementary Fig. 5 exhibit intrinsic metallic and semiconducting characteristics, respectively. The linear $I$-$V$ curves of the MoS$_2$ and graphene channel at different temperatures (Supplementary Figs. 6–7) explicitly show the good Ohmic contact for each layer at low temperatures. The low contact resistance (Supplementary Figs. 8–9) of these two separated layers undoubtedly enables us to execute both the drag and active-layer transport measurements.

Figure 2a–b illustrate the mapping of longitudinal channel resistance of graphene ($R_{Gr}$) and MoS$_2$ ($R_{MoS_2}$) in the same $V_{bg}$-$V_{tg}$ space at $T$ = 200 K in the drag device sample-S21, respectively. In general, as shown in Fig. 2a, $R_{Gr}$ is in agreement with the previous observation in a standard dual-gated monolayer graphene device[32]. However, the charge neutral resistive peak of graphene is partially screened by MoS$_2$ due to the existence of relatively high carrier density in the latter layer, yielding a weak $V_{tg}$ dependence of $R_{Gr}$ at $V_{tg}$ larger than ~ 1 V. Meanwhile, the band edge of MoS$_2$ in Fig. 2b is squeezed and held almost constant at positive $V_{tg}$, which is likely due to the contact part of the MoS$_2$ is not gated by the same gate as its major channel. Notice that the band edge of semiconducting MoS$_2$ in Fig. 2b is highlighted by green dashed line, which is quantitatively extracted from the phase signal in the lock-in measurement, shown in Supplementary Fig. 10.

As a consequence, limited by the screening effect and the contact barriers, the drag response in the current device is confined to the electron-electron regime, making hole drag not accessible. For the drag measurements, we passed a drive current ($I_{drive}$) through the active layer and measured the resulting voltage drop ($V_{drag}$) across the passive layer under open-circuit conditions. To eliminate spurious drag signals in the passive layer caused by drive-bias-induced AC gating effects[33], we have adopted a balance-bridge setup (comparison between lock-in measurement and the bridge methods can be seen Supplementary Fig. 11)[10]. Figure 2c shows the drag resistance $R_{drag}$ using graphene as the driving layer (i. e., the active layer). Here, the parasitic signals, determined to coincide with the band edge of MoS$_2$ by the phase measurements in Supplementary Figs. 10 and 12, are blanked for visual clarity. The PADR, defined as $I_{drag}/I_{drive}$ = $R_{drag}/R_{passive\ layer}$, is usually a direct measure of the interlayer interaction in drag systems. For example, when it comes to a perfect drag in the scenario of exciton condensation, PADR may reach the unity[12,34,35]. In our system, PADR (Fig. 2d) has a maximum value of ~ 0.6 when MoS$_2$ serves as the active layer, much higher compared to most of the conventional drag systems. This large PADR likely originates from more efficient momentum transfer, which is induced by strong interlayer correlation coupled with weak Coulomb screening within the 2D semiconductor MoS$_2$[19,36,37]. Moreover, we notice that maximum $R_{drag}$ seems to take place at the onset of the semiconducting MoS$_2$ channel conductance derivative with respect to gate voltage ($dG/dV_g$), as indicated by the red arrows in Fig. 2c–d (also discussed in

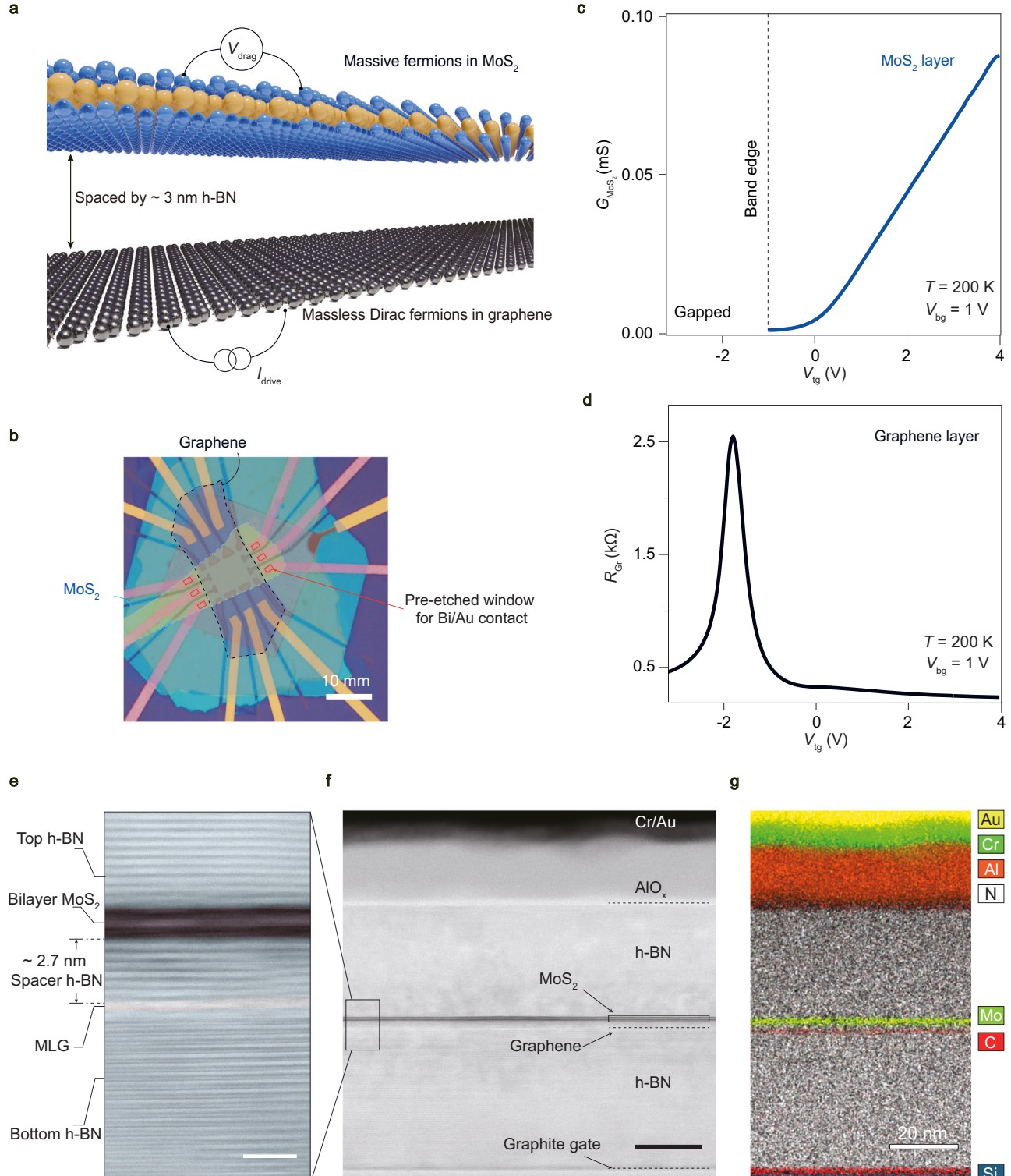

**Fig. 1 | Coulomb drag responses in a semiconductor-semimetal hybrid.**
**a** Cartoon drawings of the massive-massless Coulomb drag realized in a MoS$_2$-graphene heterostructure separated by an ultrathin 3 nm hexagonal boron nitride (h-BN) dielectric. **b** Optical image of a typical MoS$_2$-graphene drag device (sample-S21, bilayer MoS$_2$ is used as the semiconducting channel). **c, d** Line profile of field-effect curves, showing the conductance of MoS$_2$ ($G_{MoS_2}$) and the resistance of graphene ($R_{Gr}$) as a function of top-gate voltage ($V_{tg}$), measured in each constituent layer at the bottom gate $V_{bg} = 1$ V, and $T = 200$ K. **e** Zoomed-in bright-field scanning

transmission electron microscopy (BF-STEM) image of the boxed region in (**f**). It is seen that the bilayer MoS$_2$ and monolayer graphene (MLG) are separated by a thin h-BN layer. The scale bar is 2 nm. **f, g** BF-STEM image and electron energy loss spectroscopy (EELS) mapping of the cross section of a typical drag device. The scale bars in (**f**)–(**g**) are 20 nm. The black box in the right panel of (**f**) indicates the position of MoS$_2$ channel, while the dashed lines serve as guides for the eyes, marking the relative positions of the Cr/Au contact, AlO$_x$ dielectric layer, graphene channel, and graphite bottom gate.

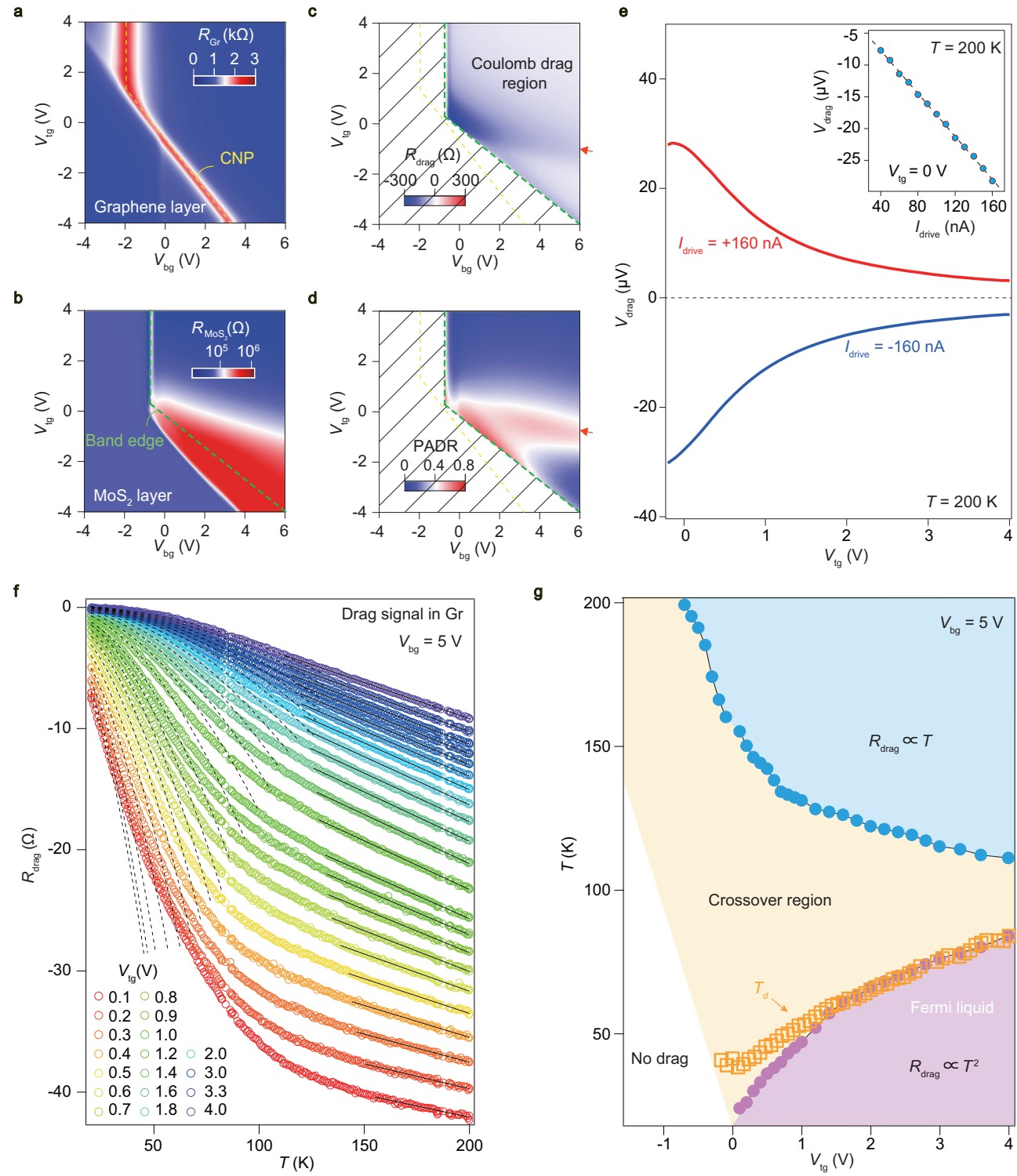

Supplementary Fig. 13). Line profiles of $R_{drag}$ in different directions of drive current ($\pm 160$ nA) as a function of $V_{tg}$ at $V_{bg} = 5$ V and $T = 200$ K are shown in Fig. 2e. The inset in Fig. 2e shows the extracted $V_{drag}$ at $V_{tg} = 0$ V as a function of $I_{drive}$. The linear relationship between $V_{drag}$ and $I_{drive}$, together with the interlayer leakage current (Supplementary Fig. 14), were carefully checked to confirm the validity of our drag measurements. Furthermore, this observed linear response of $V_{drag}$ to $I_{drive}$ persists at low temperature (as shown in Supplementary Fig. 15), suggesting that the measured drag signal is reliable under these experimental conditions, and the Onsager reciprocity when the drive and drag layers are exchanged (Supplementary Fig. 16) also

demonstrates that the system is in the linear response regime, allowing the extraction of the drag resistance from the slope of $\Delta V_{drag}/\Delta I_{drive}$.

## Temperature dependences of the Coulomb drag in the semiconductor-semimetal hybrid

In the following, we investigate the observed drag response in the graphene-$MoS_2$ hybrid at different temperatures. When lowering the temperature, phonon scattering in the $MoS_2$ is known to be largely suppressed and the carrier transport in the system is supposed to be driven from phonon-limited low-mobility regime into the intrinsic high-mobility regime, with the system exhibiting a transition from

**Fig. 2 | Gate- and temperature-dependence of the drag responses.**
**a** Longitudinal resistance ($R_{xx}$) mapping in the $V_{tg}$-$V_{bg}$ space of the graphene channel ($R_{Gr}$) in sample-S21. The yellow dashed line in (a) indicates the positon of charge neutrality point (CNP) in the graphene channel. **b** $R_{xx}$ mapping in the $V_{tg}$-$V_{bg}$ space of the MoS$_2$ channel ($R_{MoS_2}$). Data obtained at $T$ = 200 K and $B$ = 0 T. **c** Drag resistance ($R_{drag}$) in the same device. **d** Passive-to-active drag ratio (PADR) for the drag signal tested in the MoS$_2$ layer. Notice that a portion of the map in (**b**) and (**d**) are masked (ill defined signal since the lock-in amplifier is out of phase, as seen in Supplementary Fig. 12), for visual clarity. The dashed green lines in (**b**)–(**d**) indicate the band edge of the MoS$_2$ channel. The red arrows in (c) and (d) mark the positions where the maximum value of $R_{drag}$ occurs at $V_{bg}$ = 6 V. **e** The nearly symmetric drag response in different directions of drive current at $T$ = 200 K. The red and blue curves represent the drag voltage ($V_{drag}$) as a function of drive current ($I_{drive}$) for opposite directions, the magnitude of drive current is 160 nA. The inset shows the extracted values of $V_{drag}$ at $V_{bg}$ = 1 V as a function of $I_{drive}$. The black dashed line represents the linear fit, demonstrating the linear response of the drag signal to $I_{drive}$. **f** Temperature dependence of $R_{drag}$ (colored open symbols) at $V_{bg}$ = 5 V for different $V_{tg}$. The black dashed lines represent the fits to the low-temperature data with a quadratic temperature dependence, while the black solid lines correspond to the fits to the high-temperature data, assuming a linear temperature dependence. **g** The blue, yellow, and purple filled areas show the $R_{drag}(T)$ responses with $T$-linear, $T$-$T^2$ crossover, and $T^2$ behavior, respectively. Boundaries (blue and purple solid circles) are obtained by fitting the lines in (**f**). The open square symbols are the critical temperature $T_d$, which is defined as $T_d = E/k_B k_F d$ with $k_B$, $k_F$ and $d$ being the Boltzmann constant, the magnitude of Fermi wavevector, and the interlayer distance, respectively.

insulating behavior to a metallic one, known as metal-insulator transition (MIT) when varying from low to high carrier density in the low temperature limit[31,38,39]. This highly tunable electron transport properties in one of the layers of the drag system may give rise to unique and unconventional drag signals, distinguishing it from previously reported drag systems[21,37,40,41].

Figure 2f shows the temperature dependence of drag resistance at different $V_{tg}$ with the $V_{bg}$ fixed at 5 V. $R_{drag}$ increases monotonically as the temperature increases when MoS$_2$ becomes metallic at large $V_{tg}$. A $T^2$ dependence is clearly observed at the base temperatures, which is in good agreement with the theory of frictional drag for Fermi liquid[42,43]. However, in the high-temperature regime, deviation from the $T^2$ dependence becomes pronounced, eventually evolving into a linear temperature dependence. It is found that the crossover regime from $T^2$ to $T$ dependence broadens as MoS$_2$ becomes more insulating with decreasing $V_{tg}$.

We further plot the color maps of the drag resistance in the $T$-$V_{tg}$ phase diagram, as shown in Fig. 2g. The corresponding fitting points for $R_{drag}$ in Fig. 2f are featured in the phase diagram. From this comparison, we identify four distinct temperature-dependent drag regions: no drag, $T^2$, $T$ and $T^2$-$T$ crossover regions are observed. Coulomb drag resistance is known to be extremely sensitive to temperature, interlayer spacing, carrier density (or density mismatch between the layers) and magnetic field[22,44]. And drag transport regimes can be defined by the Fermi energy $E_F$, the magnitude of Fermi wavevector $k_F$, interlayer separation $d$. In the Boltzmann-Langevin theory of Coulomb drag for the Fermi liquid scenario, at low temperatures ($T \ll T_d = E_F/k_F d$) and in the clean limit (weak disorder or low scattering rate), drag is dominated by the particle-hole continuum and $R_{drag}$ is proportional to $T^2$[45]. Thus, we have plotted the estimated characteristic temperature $T_d$ in the phase diagram and found that the curve of $T_d$ indeed separates the quadratic $T^2$ and linear $T$ dependent drag regimes. The temperature region of a Fermi liquid below $T_d$, in which the drag resistance follows the $T^2$ law, is strongly suppressed as the band edge of MoS$_2$ is approached. At higher temperatures, $T > T_d$, phase-space constraints due to small-angle scattering lead to a linear temperature dependence[2,45]. While on the insulating side, $R_{drag}$ deviates from both $T^2$ and linear temperature dependence, eventually drops to zero as the carrier density in MoS$_2$ decreases and Fermi level moves into the bandgap of MoS$_2$ (marked as "no drag" in the diagram).

**Drag at the matched density**
It is noticed that the carrier density dependent characteristic of $R_{drag}$ varies significantly in different kinds of drag system. The relationship between drag resistance and carrier density at the matched density in massive-massless fermion system has been explored theoretically, which is in contrast with that in massive-massive and massless-massless fermion systems[25,46]. For high density regime ($k_F d \gg 1$), all three systems exhibit a similar carrier density dependence, specifically following an $1/n^3$ behavior. For the low density regime ($k_F d \ll 1$), the carrier density dependence exhibits distinct characteristics for different systems, highlighting their unique properties. Specifically, in the massless-massive case, $R_{drag}$ scales as $1/n^2$, whereas for massive-massive and massless-massless systems, the dependencies are predicted to follow $1/n^3$ and $1/n$, respectively[25,43].

In our case, we first estimate the carrier density of the MoS$_2$ and graphene layer ($n_{MoS_2}$ and $n_{Gr}$, respectively) independently from the longitudinal and transverse resistance ($R_{xx}$ and $R_{xy}$, respectively) of the two layers based on measurements of the classical Hall effect at 77 K, as shown in Fig. 3a and b. The $R_{xy}$ of both MoS$_2$ and graphene varies linearly with the magnetic field, as shown in Supplementary Fig. 17. The equal density line ($n_{MoS_2} = n_{Gr}$) can be easily identified by subtracting the two carrier density color maps of MoS$_2$ and graphene, as illustrated by the black dash line in Fig. 3c. Subsequently, we plot the drag resistance along the density matched line ($n_{MoS_2} = n_{Gr}$) in logarithmic scale and converges to the expected $1/n^\alpha$ dependence, with $\alpha \approx 2$. For our massive-massless fermion system, the range of the equal density line is about 1.2 ~ 3.0 × 10$^{12}$ cm$^{-2}$. Thus, we estimate that the maximum value of $k_F d$ satisfies $k_F d < 1$ using the expression $k_F = \sqrt{\pi n}$. This result demonstrates the equal density line is at the low density regime with the $1/n^2$ dependent $R_{drag}$, which agrees well with the theoretical predicts[25]. Our analysis on the matched-density drag response suggests that, although being at a temperature slightly above $T_d$, the drag response can still be considered as a Fermi liquid at proper doping.

**Asymmetric drag as a compressibility spectrometer at high temperature**
For a better understanding the charge transport in this asymmetric drag system, we first extracted the Hall carrier density as a function of temperatures and gate voltages, and calculated the Hall mobility for each layer accordingly (Supplementary Fig. 18). The field-effect mobility obtained from the transfer curves of these two layers is also presented in Supplementary Fig. 19 for comparison. We now show the magnetodrag (the longitudinal component) of massive-massless fermion system in the presence of a finite magnetic field $B$. The Onsager reciprocity relation for magnetodrag resistance remains valid regardless of whether the system is within or outside the Fermi liquid regime (as shown in Supplementary Figs. 20 and 21). It is noteworthy that a slight departure from perfect reciprocity occurs at low temperatures and under magnetic fields. While the microscopic origins of this non-ideal reciprocal Coulomb drag remain unclear, they may stem from a high concentration of disorder in specific devices, which can introduce a small rectified component to the drag signal. Subsequently, we investigate the $B$-dependent $R_{drag}$ at liquid nitrogen temperature ($T$ = 77 K). It is clearly seen in Fig. 4a that MoS$_2$ is absent of magneto-resistance, as it is known to exhibit negligible magneto-resistance and hence flat-compressibility due to the low mobility shown in Supplementary Figs. 18 and 19 at high temperatures[47]. Figure 4b illustrates the magnetoresistance of the graphene channel itself in the parameter space of $B$-$V_{tg}$. The SdH oscillations within ± 12 T for graphene is essentially featureless at this temperature, which is in agreement with literatures[48,49]. Remarkably, magnetodrag, i.e., $R_{drag,xx}$-$B$ in Fig. 4c

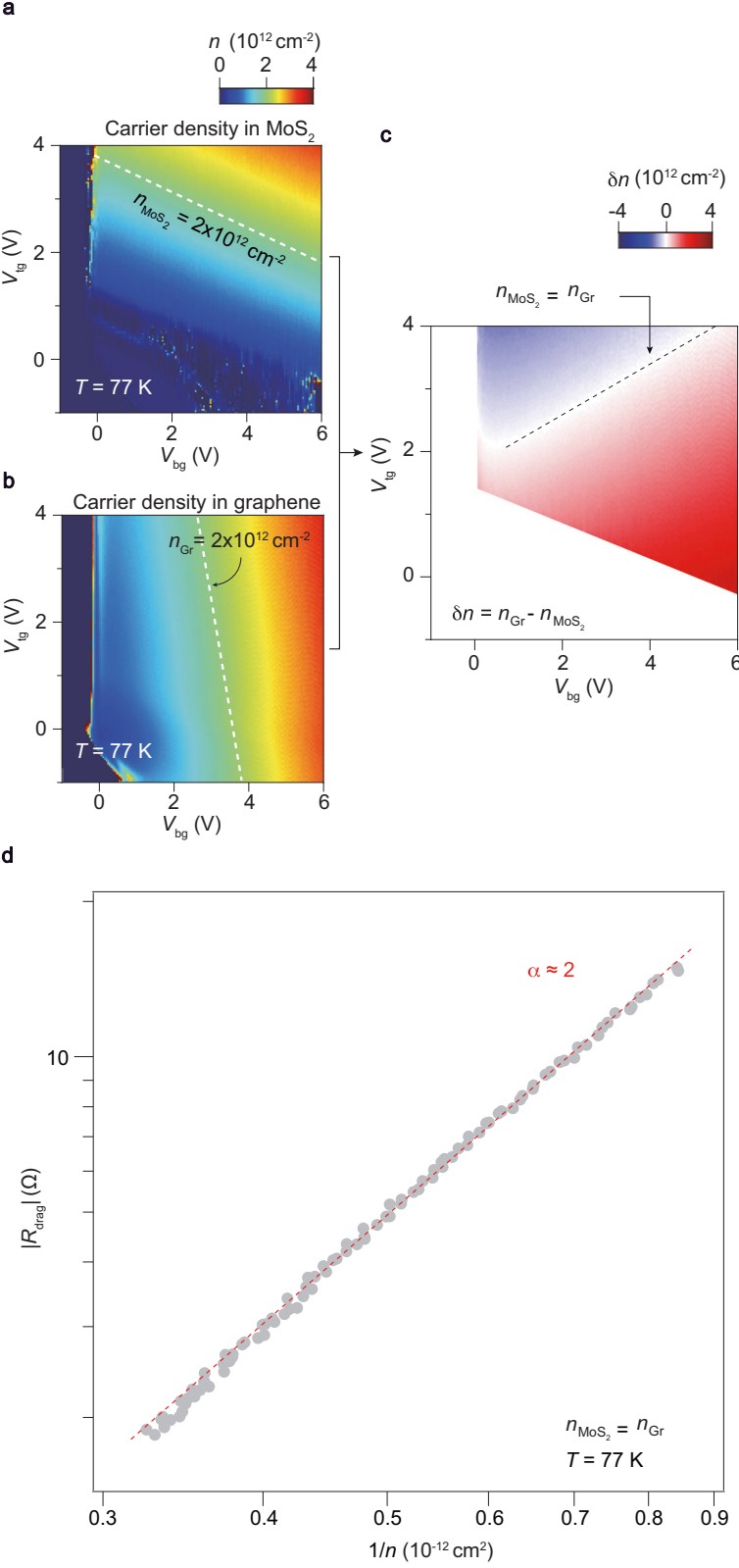

**Fig. 3 | Drag at the matched density. a, b** The carrier density $n$ of the MoS$_2$ layer ($n_{MoS_2}$) and graphene layer ($n_{Gr}$) in the drag system as a function of $V_{tg}$ and $V_{bg}$. Data are obtained by using the formula $n = B/eR_H$, where $e$, $B$ and $R_H$ represent the elemental charge, out-of-plane magnetic field and Hall coefficient, respectively. $B/R_H$ is obtained by extracting the slope of Hall resistance at $B = 1$ T and 0 T. The temperature is fixed at 77 K. **c** The differential carrier density $\delta n$ plotted by subtracting the colour map (**b**) with (**a**). Notice that black dashed line indicates the scenario of matched-density between the graphene and MoS$_2$ layer. **d** $R_{drag}$ plotted alongside the black dashed line in (**c**), which shows $1/n^2$ dependence in the matched-density drag, suggesting the validity of the Fermi-liquid picture. The red dash line indicates the power-law fit, $R_{drag} \propto 1/n^{\alpha}$, where $\alpha$ is the power-law exponent.

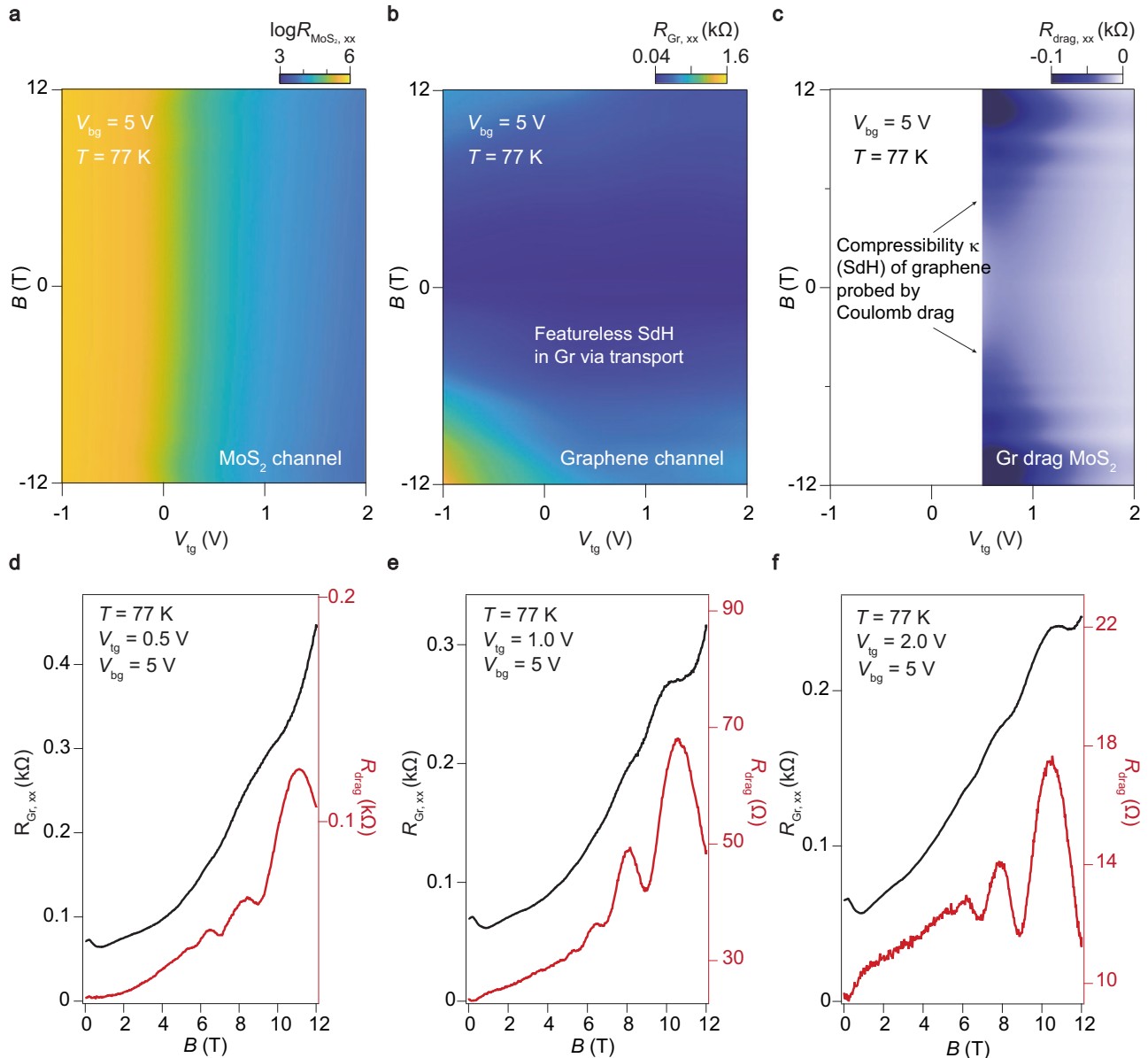

**Fig. 4 | Quantum-oscillations in MoS₂-graphene drag system at different gate voltages at 77 K. a, b** Channel resistance as a function of $V_{tg}$ and out-of-plane magnetic field $B$, recorded by standard lock-in transport for MoS₂ and graphene, respectively. Data were measured at $T = 77$ K and $V_{bg} = 5$ V. It is seen that neither MoS₂ nor graphene exhibits visible quantum oscillations via transport. **c** Clear

Shubnikov-de Haas (SdH) features captured by Coulomb drag under the same experimental conditions. (**d**)-(**f**) Line profiles of $R_{Gr}$ (solid black lines) and $R_{drag}$ (solid red lines) as a function of magnetic field at $T = 77$ K and $V_{bg} = 5$ V for $V_{tg} = 0.5$ V, 1.0 V and 2.0 V, respectively.

shows well-developed stripped features (indicated by the arrows), which are the quantum oscillations from graphene probed, with a significantly amplified amplitude, via Coulomb drag.

According to the standard perturbative theory, the phenomenological drag resistivity $\rho_D$ can be written as[1,43]

$$\rho_D = \frac{-1}{16\pi S k_B T \sigma_1 \sigma_2}$$
$$\times \sum_q \int_{-\infty}^{\infty} d\omega \frac{|U_{12}(q,\omega)|^2 Im\Pi_1(q,\omega) Im\Pi_2(q,\omega)}{\sinh^2(\hbar\omega/2k_B T)}, \tag{1}$$

where $S$ is the cross section area of the layers; $\sigma_1$ and $\sigma_2$ denote the Drude conductivities of each layer; $U_{12}$ is the screened interlayer interaction; $\Pi_i$ is the density-density response of layer $i$. The magnetic-field dependence at low $q$, $\omega$ enters through the layers' compressibilities via $Im\Pi_i \propto \nu_i(B)\omega/(\nu_{Fi}q)$. $\nu_i$ is the thermodynamic density of

states DOS (compressibility $\kappa = dn/d\mu$, is proportional to DOS); $\nu_{Fi}$ is the Fermi velocity; $d$ is the interlayer spacing; dominant momenta satisfy $q \lesssim q_T \equiv k_B T/\hbar\nu_{Fi}$.

In a MoS₂/graphene bilayer with a 3 nm h-BN spacer, MoS₂ has thermally smeared Landau quantization at 100 K (as the cyclotron energy of MoS₂ at 12 T is only 2-3 meV, way smaller than $k_B T$ ~ 8.6 meV), giving nearly constant $\nu_2(B)$, whereas graphene retains oscillatory $\nu_1(B)$ governed by Lifshitz-Kosevich/Dingle factors. Consequently,

$$\delta\rho_D(B) \propto T^2 \, \delta\nu_1(B) \times \mathcal{S}(B), \tag{2}$$

where thermal and disorder damping are encoded in $R_T$ and $R_D$ inside $\delta\nu_1(B)$, and $\mathcal{S}(B)$ denotes a weakly $B$-dependent screening factor. Therefore, in this current scenario, Coulomb drag functions as a compressibility transducer of graphene's SdH oscillations with a growing $T^2$ baseline, and more importantly, an interaction-induced

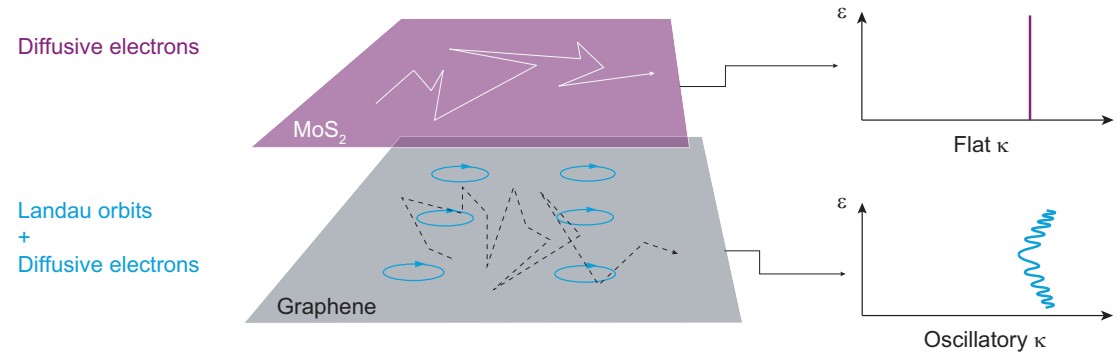

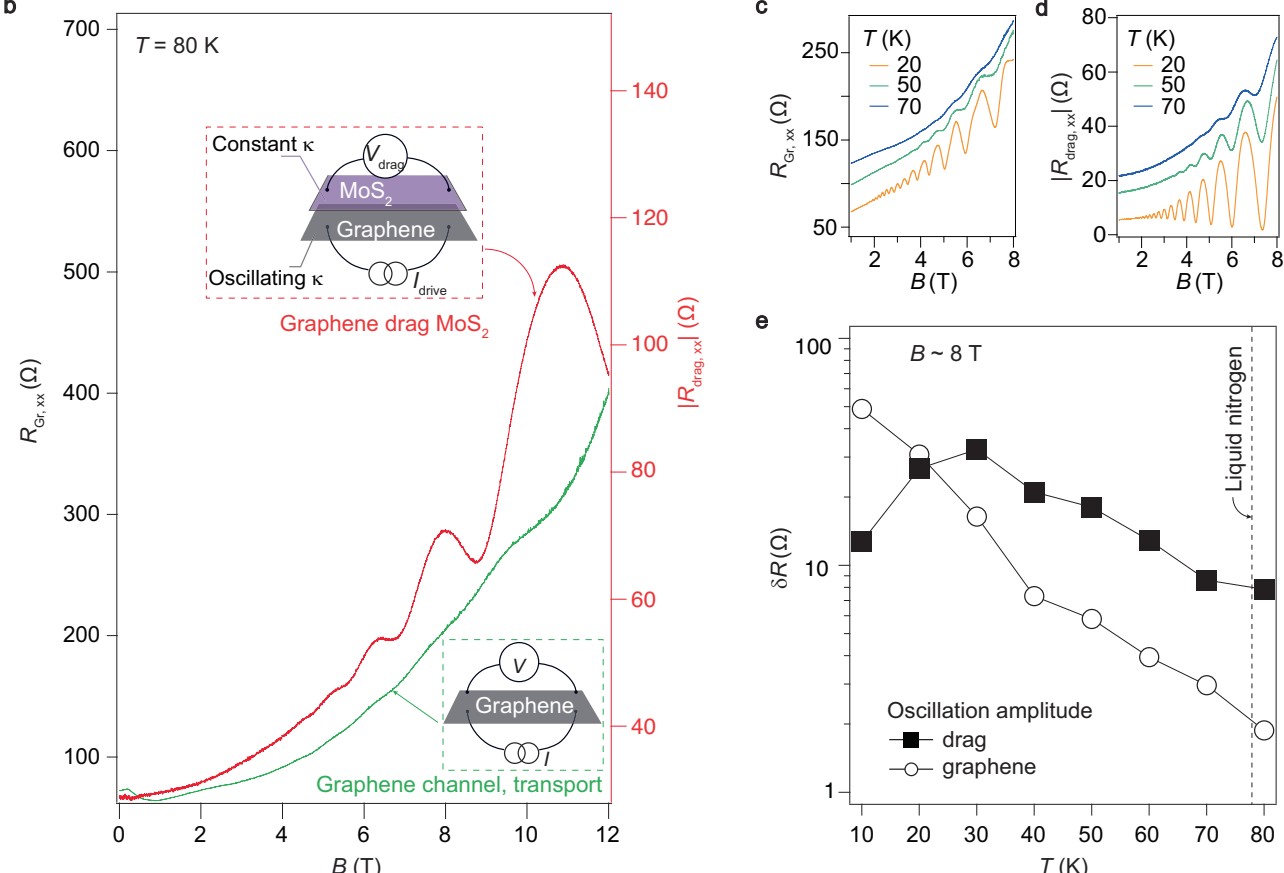

**Fig. 5 | Drag-boosted quantum-oscillation visibility in graphene at elevated temperature. a** The schematic illustration of diffusive conductive model in low-mobility $MoS_2$ layer with flat electron compressibility and quantized electron states in high-mobility graphene layer with oscillatory electron compressibility. In the right panel of (**a**), the axes $\epsilon$ and $\kappa$ represent energy and electron compressibility, respectively. **b** Line profiles of $R_{Gr}$ (green) and $R_{drag}$ (red) as a function of magnetic field, respectively. Insets show the cartoon illustrations for each measured structure. **c, d** Quantum oscillations obtained by transport in graphene channel and by Coulomb drag, respectively, while $V_{tg}$ and $V_{bg}$ are fixed at ~1 V and 5 V, respectively. The orange green and blues lines represent the curves measured at 20, 50 and 70 K, respectively. **e** The extracted amplitudes of quantum oscillations captured by transport in graphene (open circles), and by Coulomb drag (solid squares), respectively. It is seen that the drag plays a role of transducer of electron compressibility (which is seen as SdH oscillations) in graphene, and amplifies the quantum oscillations at elevated temperatures. Above liquid nitrogen temperature, the amplitude can be amplified by over one order of magnitude.

factor of $|U_{12}|^2$ allows clear readout of oscillations at 100 K even when graphene's own magnetotransport oscillations are muted. The strong interlayer coupling at 3 nm (dominant $q \sim k_B T/\hbar v_F$ with $qd \approx 0.04 \ll 1$) prevents exponential suppression over spatial distance, and physically, drag tracks the oscillatory compressibility of graphene as

$$\delta\rho_D \propto T^2 |U_{12}|^2 \delta\kappa_{graphene}(B). \quad (3)$$

We firstly compare the intrinsic magneto-resistance of the graphene layer with magneto-drag response for various $V_{tg}$ by fixing the temperature and back gate, which is shown in Fig. 4d–f. As $V_{tg}$ increases, both the values of $R_{xx}$ of graphene and drag drop due to more carriers enter into the channel. Meanwhile, the increased conductivity of the $MoS_2$ layer gives rise to the proximity screening effect, which in turn effectively suppresses charge inhomogeneity in the graphene layer[50]. Therefore, the electron mobility in graphene will be

greatly enhanced (Supplementary Fig. 18), leading to more evident quantum oscillations. However, when $V_{tg}$ decreases, the weaken proximity screening effect predictably enhances the electron-electron interactions and $|U_{12}|^2$, amplifying the amplitude of the oscillations in drag signal.

Subsequently, we studied the temperature dependence of the magneto-drag resistance. Here, we included an intuitive schematic illustration in Fig. 5a for better understanding. The low-mobility electrons in $MoS_2$ move randomly between scattering disorders with a magnetic field applied (in the top layer). This diffusive transport model prevents the observation of significant SdH oscillations under 12 T, meaning the electron compressibility in $MoS_2$ is nearly flat as a function of the magnetic field. In stark contrast, the high-mobility electrons in graphene primarily follow cyclotron orbits perpendicular to the magnetic field (in the bottom layer). In other words, the electron states could be easily quantized into distinct Landau Levels. Consequently, the electron compressibility in graphene oscillates with the magnetic field for a fixed carrier density, allowing for the observation of evident SdH oscillations. However, as the temperature increases, transport across the sample occurs via inter-orbit scattering (the dashed lines connecting the orbits), which is a diffusive process, leading to the amplitude of the SdH oscillations to decrease until they vanish.

Indeed, as shown in Fig. 5b, at $T$ = 80 K, it is clearly seen that the transport channel of graphene's resistance (green) has essentially featureless SdH oscillations, with only a few bumps difficult to distinguish. But for the case of drag system, the Coulomb drag response (red) still exhibits significant oscillations, agreeing well with the above analysis. Figure 5c and d further show the side-by-side comparison of oscillations in lateral transport and drag response at different temperatures. Notice that these drag oscillations versus $1/B$ are spaced exactly like graphene's SdH oscillations, as shown in Supplementary Fig. 22. In addition, the flat electron compressibility in $MoS_2$ confirms the origin of the observed oscillations in $R_{drag}$ is solely from the graphene layer, suggesting that $MoS_2$ serves as an excellent sensor of probing electron compressibility in high-mobility materials. Figure 5e shows the extracted oscillation amplitudes obtained from lateral transport in graphene (open circles) and from Coulomb drag (solid squares), after subtracting the background from each curve. It is seen that at base temperature lower than 20 K, the SdH oscillations measured from lateral transport in graphene has larger amplitudes than those from the drag response. At elevated temperatures, the amplitudes of drag oscillation rapidly overgrow, functioning as a transducer of $\kappa_{graphene}$ that effectively amplifies the readout of quantum oscillations. Above liquid nitrogen temperature (indicated by the dashed line in Fig. 5e), the oscillation amplitude difference can be more than one order of magnitude. This kind of result is reproducible in sample-S24, which featured the same device geometry as sample-S21, as shown in Supplementary Fig. 23.

The final section addresses the dependence of the drag response on the spacer thickness $d$, a key factor determining the strength of interlayer coupling within the drag system. We have measured the dual-gate mapping of drag resistance for three typical samples with different interlayer spacing, as shown in Supplementary Fig. 24a–c. We could clearly see that the drag response is reproducible in different samples, and the magnitude of drag resistance is inversely proportional to $d$, which is consistent with previous reports[3,27]. For better comparison, we plotted the drag resistance as a function of carrier density for samples-S21 and S15 at 77 K in Supplementary Fig. 24d. Moreover, we utilized drag and transport measurements at 77 K to probe the electron compressibility of graphene in two drag systems with different $d$, as presented in Supplementary Fig. 24 (e). The carrier densities for this measurement were fixed at $\sim 1.2 \times 10^{12} cm^{-2}$ for $MoS_2$ and $\sim 2.7 \times 10^{12} cm^{-2}$ for graphene, respectively. The system with the thinner spacer (sample-S21) exhibited stronger interlayer coupling, resulting in a higher magnitude of oscillations in the drag signal

compared to the weaker interlayer coupling of the thicker spacer (sample-S15). This significant difference indicates that the thickness of the spacer is critically important for efficiently probing compressibility in the drag system. We have summarized the characteristics of temperature, magnetic field, and carrier density dependence in a collection of experimentally tested Coulomb drag systems[3,13–15,18,21,22,27,34,40,51–53] in Table 1. The unique role of $MoS_2$ in the observation of pronounced SdH oscillations is of two origins, i.e., relatively flat electron compressibility in its own, and sufficiently strong Coulomb interaction factor $|U_{12}|^2$. In this regard, if $MoS_2$ in the current system is replaced by other 2D semiconductor with flat electron compressibility - such as $MoSe_2$ and $WSe_2$, provided that Ohmic contacts could be obtained at low temperatures - should theoretically exhibit similar phenomena. Otherwise, for instance, replacing $MoS_2$ with graphene, high screening in both layers will lead to a dramatic reduction of interlayer Coulomb interaction, and their SdH signals tend to be entangled, complicating the decoupling of layer-specific information. Overall, the massless Dirac -massive Schrödinger fermions graphene-$MoS_2$ drag system in this work demonstrates a distinct paradigm where the asymmetric configuration in each constituent layer can lead to selective-&-amplified probe of electron compressibility, thus providing grounds for future investigations in drag electronics.

To conclude, by integrating an Ohmic-contacted semiconducting TMD channel with a semimetal, we realize a strongly coupled graphene/$MoS_2$ bilayer separated by an ultrathin $\sim 3$ nm h-BN spacer that enables interaction-assisted probe of quantum thermodynamics. Systematic maps of the drag response $R_{drag}$ versus temperature, carrier density, and magnetic field place our device in the degenerate Fermi-liquid regime (for $T \lesssim 100$ K with $MoS_2$ in its conduction band). Crucially, the $MoS_2$ layer's essentially flat compressibility makes it an efficient transducer of graphene's Landau quantization. At elevated temperatures and moderate fields, the drag signal exhibits clear SdH-like oscillations that persist even when graphene's own magnetotransport is featureless. The oscillation periodicity of the magnetodrag response follows graphene's Landau quantization, consistent with a Fermi-liquid perturbative picture in which $\delta R_{drag} \propto T^2|U_{12}|^2\delta\kappa_{graphene}(B)$. Beyond extending the drag landscape to mixed massive-massless double layers, these results establish Coulomb drag as a practical compressibility spectroscopy for 2D materials, unlocking access to quantum phenomena that can be suppressed in standard transport. More broadly, the semiconductor-semimetal architecture suggests a route to interaction-engineered sensors and high-temperature quantum-oscillation metrology, with tunable sensitivity via spacer thickness, screening environment, and carrier density.

## Methods
### Sample fabrication
vdW few-layers of the h-BN/$MoS_2$/h-BN/graphene/h-BN sandwich were obtained by mechanically exfoliating high quality bulk crystals. The vertical assembly of vdW layered compounds was fabricated using the dry-transfer method in a nitrogen-filled glove box. The heterostructures were then transferred onto the pre-fabricated Au or graphite gates. Hall bars of the devices were achieved by reactive ion etching. During the fabrication processes, electron beam lithography was done using a Zeiss Sigma 300 SEM with a Raith Elphy Quantum graphic writer. One-dimensional edge contacts of monolayer graphene were achieved by using the electron beam evaporation with Ti/Au thicknesses of $\sim 5/50$ nm and the window contacts of bilayer $MoS_2$ were fabricated with a thermal evaporator, with typical Bi/Au thicknesses of $\sim 25/30$ nm. After atomic layer deposition of about 20 nm $Al_2O_3$, big top gate was deposited to form the complete dual gated h-BN encapsulated drag devices as shown in Fig. 1a and b. Note that we always put 2D semiconductor $MoS_2$ on the top of graphene because we adopted window-contact technique as previously reported[31]. We firstly

**Table 1 | A summary of the characteristics for different drag systems**

| Drag category | Drag system | $T$ dependence | $n$ dependence | $B$ dependence | Maximum drag resistance | Ref. |
|---|---|---|---|---|---|---|
| Massless-massless fermions | ML Gr-ML Gr | $T^2$ (high density) | N/A | anomalous | 50 Ω | 3 |
| | ML Gr-ML Gr | $T^2$ (0 T) | N/A | $B^2$ | 400 Ω (70 K, 1 T at CNP) | 51 |
| Massless-massive fermions | ML Gr-Carbon nanotube | $T$ (when $T > T_F$) | $1/(V_g - V_0)^{1 \sim 2}$ | N/A | 6 Ω (260 K) | 15 |
| | ML Gr-InAs nanowire | $T^2$ | $1/n^4$ | $B^2$ | 0.5 Ω (1.5 K) | 18 |
| | ML Gr-GaAs 2DEG | $T^2 \lg T$ | N/A | N/A | 2 Ω (0.24 K) | 27 |
| | ML Gr-BL Gr | $T^2$ (high density) | $1/n^2$ (low density), $1/n^3$ (high density) | N/A | 5 Ω (high density), 50 Ω (CNP) | 52 |
| | ML Gr-BL MoS$_2$ | $T^2 \sim T$ | $1/n^2$ | $B$ | 0.3 kΩ (200 K, 0 T) | This work |
| Massive-massive fermions | BL Gr-BL Gr | $R_{drag}$ decreases as $T$ increases | N/A | N/A | 800 Ω (1.5 K at CNP) | 21 |
| | BL Gr-BL Gr | $T^2$ (nonlocal), $T^4$ (local) | $1/n^3$ (nonlocal, low density) | N/A | 60 Ω (CNP) | 22 |
| | ML MoSe$_2$-ML WSe$_2$ | $T^2$ (<10 K) | $1/(n^p - n^m)^3$ | N/A | 1 MΩ (1.5 K) | 34 |
| | FL MoS$_2$-FL MoS$_2$ | $T^2 \ln T$ | N/A | N/A | 2.5 MΩ (1.5 K) | 37 |
| | BL Gr-GaAs 2DEG | N/A | $1/n^3$ (high density) | N/A | 2 Ω (70 K) | 53 |
| Others | ML Gr-LAO/STO | $T_c \sim 0.2$ K | N/A | N/A | 0.5 Ω (0.2 K) | 13 |
| | InAs-GaSb topological wires | $R_{drag}$ decreases as $T$ increases | N/A | N/A | 0.8 kΩ (0.3 K) | 14 |
| | ML Gr-Gr/h-BN moiré superlattice | $T^2$ (high density) | $1/n^{1.3-1.7}$ (high density) | N/A | 10 Ω (CNP) | 40 |

ML, BL, and FL: monolayer, bilayer, and few-layer; Gr: graphene; 2DEG: two-dimensional electron gas; CNP: charge neutrality point of graphene.

etched through the windows on top h-BN and use it to pick up MoS$_2$ and other layers afterwards. Then, we adopted Bismuth with low work function as electrodes contacted to MoS$_2$ and achieved Ohmic contact at low carrier density and temperatures. Although the fabrication would be more challenging if the MoS$_2$ and graphene layers were swapped, the drag response would remain identical due to the vertically mirror-symmetric configuration.

### Drag measurements
In lock-in measurements, current is typically driven by applying an AC bias voltage $V_{drive}$ to one side of the channel while the other side is grounded. However, in Coulomb drag measurements, applying this bias to the drive layer may induce spurious drag signals in the drag layer due to the AC gating effect caused by the drive bias. Here we applied about 0.2 ~ 0.3 V AC bias voltage at 17.777 Hz to drive the active layer through a 1:1 voltage transformer. The transformer was connected to a 10 kΩ potentiometer, which can help to distribute the AC voltage across both ends of the driving layer. This config-uration minimizes the AC interlayer potential difference in the channel, thereby reducing the AC coupling between the active and passive layers. We used two 1 MΩ resistors connected with the driving layer and measured the voltage drop across one of the resistors to obtain the driving current. The drag voltages were recorded using low-frequency SR830 lock-in amplifiers. Since the relationship between $V_{drag}$ and $I_{drive}$ is linear, we simply employed a large drive current, such as 100 - 200 nA, to improve the signal-to-noise ratio in our drag measurements without any other specific intentions. Four-probe measurements were used throughout the transport measurements in an Oxford Teslatron cryostat. Gate voltages on the as-prepared devices were controlled by a Keithley 2400 source meter.

### Data availability
The Source Data underlying the figures of this study are available at https://doi.org/10.5281/zenodo.18300086. All raw data generated during the current study are available from the corresponding authors upon request.

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

## Acknowledgements

This work is supported by theNational Key R&D Program of China (Grant No. 2024YFA1410400) and the National Natural Science Foundation of China (NSFC) (Grant Nos. 92565302, 12574074, 52188101, 12450003, and 92265203). Z.H. acknowledges the support of the Fund for Shanxi "1331 Project" Key Subjects Construction, and supports from the Innovation Program for Quantum Science and Technology (Grant No. 2021ZD0302003). K.W. and T.T. acknowledge support from the JSPS KAKENHI (Grant Numbers 21H05233 and 23H02052), the CREST (JPMJCR24A5), JST and World Premier International Research Center Initiative (WPI), MEXT, Japan.

## Author contributions

S.Z., Z.H., and W.R. conceived the experiment and supervised the overall project. Y.L. and S.Z. performed the device fabrications and electrical measurements; K.Y. and H.W. contributed to electrical measurements; Q. Z and H. L performed the cross-sectional STEM characterization of the drag device; K.W. and T.T. provided high quality h-BN bulk crystals; S.Z., Y.L. and Z.H. analysed the experimental data. The manuscript was written by Z.H., S.Z. and Y.L. with discussions and inputs from all authors.

## Competing interests

The authors declare no competing interests.
