## [Transparent Peer Review file · Nature Communications]

High-temperature probe of electron compressibility via asymmetric Coulomb drag

Corresponding Author: Professor Zheng Han

Version 1:

Reviewer comments:

Reviewer #1

(Remarks to the Author)

In the manuscript, the author conducted systematic drag experiments in a graphene/MoS₂ double layer. The authors observed clear SdH-like oscillations in the drag signal that precisely track the SdH within the graphene layer. As noted by the authors, the MoS₂ layer's nearly flat compressibility facilitates the visualization of graphene's Landau quantization. While the experimental study is systematic with clear data presentation and highly readable writing, the manuscript lacks a compelling focus. The central claim in the abstract section, i.e. the observation of pronounced SdH oscillations in the drag signal, is only introduced in Figure 4. Figures 1 to 3 appear more as a simple presentation of the drag characteristics in the massless-massive system, without sufficient connection to the main finding. The authors should refine the manuscript to better highlight the key points presented in Figure 4 and strengthen the discussions on the underlying mechanisms and potential impact, which will undoubtedly improve the chances of this work being accepted.

Below are some minor comments.

1. Could the authors provide more detailed discussions on the effect of bottom gate geometry on the measured drag behaviors (Supplementary Figure 4)?
2. Have the authors checked the linear response of the drag signal to I_{drive} at low temperatures?
3. The remarkably high passive-to-active drag ratio is intriguing. Could the authors offer a possible explanation for this observation?
4. The red arrows referenced in Fig. 1g-h are missing from the actual figures.

Reviewer #2

(Remarks to the Author)

Liu et al. report compressibility spectroscopy in two-dimensional systems using an asymmetric Coulomb drag configuration, particularly in the Fermi-liquid regime. The presented results are intriguing and could serve as a powerful methodology for probing subtle quantum phenomena in graphene. However, the fundamental characteristics of the active and passive layers are not clearly defined, and the terminology employed throughout the manuscript lacks clarity and consistency. The following issues should be carefully addressed before the manuscript can be considered for further consideration.

(1) The completed hBN dual-gated device structure should be characterized using TEM to confirm the thicknesses of each layer (Al₂O₃, top and bottom hBN, hBN spacer, graphene, and MoS₂) and to assess the interface quality. In addition, a cross-sectional device image would greatly improve understanding, including explicit labeling of which layer acts as the active (drive) and passive (drag) layers in all figures.

(2) Please clearly indicate whether bilayer MoS₂ is solely used and consistently serves as the passive layer throughout the manuscript. The current description is confusing.

(3) The drain current as a function of top- and bottom-gate voltages (V_{tg} and V_{bg}) for different temperatures should be provided separately, including both transfer and output characteristics.

(4) The leakage current between the active and passive layers should be presented for all measurement conditions, particularly as a function of V_{tg} , V_{bg} , temperature, and magnetic field.

- (5) Both field-effect mobility and Hall mobility as a function of temperature should be included to provide a clearer understanding of charge transport.
- (6) The Onsager reciprocity relation should be verified and discussed to confirm the presence of genuine Coulomb drag phenomena.
- (7) The bias condition (e.g., $I = 160$ nA) should be clarified, including why this particular condition is only considered.
- (8) The contact resistance should always be specified, as it directly affects both the drag and active-layer transport measurements.
- (9) The electrical bias conditions and temperatures differ across the figures. Such variations must be discussed comprehensively before presenting the data, to ensure consistent interpretation. Limited fundamental experimental results currently make the discussion difficult to follow.
- (10) Both V_{tg} and V_{bg} values should be explicitly annotated in all subfigures (e.g., Fig. 1i).
- (11) The meaning of the y-axis in Fig. 3d should be clearly defined.
- (12) The carrier density measurements in Fig. 3 (at 100 K) and the Coulomb drag data in Fig. 4 (at different temperatures) must be correlated. The Hall mobility and carrier density should be presented as a function of both different temperature and bias for consistency.
- (13) In Fig. 1b, the scale bar indicates 10 mm, please confirm if this is a typo.
- (14) In Fig. 1c, the data points corresponding to GMoS2 for $V_{tg} = -3$ V to -1 V appear missing. In addition, including a semi-logarithmic scale would help reveal the n-type semiconducting behavior of bilayer MoS2.
- (15) In Fig. 1f, please define the parameter RMoS. In Fig. 1h, define PVR explicitly. Numerous typographical errors are present throughout the manuscript and should be corrected carefully.
- (16) In Fig. S6c, the V_{tg} -dependent MoS2 resistance behavior is nearly symmetric and resembles that of graphene, suggesting ambipolar characteristics. Please elaborate on this observation.
- (17) In Fig. S10a, GMoS2 measured at $V_{bg} = 4$ V is lower than that at $V_{bg} = 1$ V (Fig. 1c), which is counterintuitive for n-type MoS2. Please clarify this discrepancy. If different devices were used, please specify device numbers in all figures.
- (18) The thickness of the hBN spacer is a critical parameter for interlayer coupling at the Gr/hBN/MoS2 interface. The electron compressibility as a function of hBN thickness should be presented and discussed. In addition, please discuss the case where the positions of MoS2 and graphene are swapped. Why is MoS2 always placed on top of graphene? Addressing this question would strengthen the physical reasoning.
- (19) The contact resistance for both MoS2 and graphene as a function of bias and temperature should be presented to ensure accurate analysis of interlayer transport behavior.
- (20) Since the effective mass of MoS2 is roughly an order of magnitude larger than that of graphene, it would be helpful to include an intuitive schematic illustration showing how the Shubnikov–de Haas oscillations in MoS2 could be amplified or modulated through electron compressibility.

In summary, while the manuscript presents an interesting approach to probing compressibility in two-dimensional systems via Coulomb drag, it currently lacks structural, electrical, and conceptual consistency. The figures require systematic reorganization, and several critical parameters (bias, temperature, and layer identity) must be clarified before the study can provide a coherent physical picture. If the manuscript provides a more detailed discussion of the fundamental charge transport characteristics in each layer, a deeper and more quantitative review of the Coulomb drag phenomena would be possible.

Reviewer #3

(Remarks to the Author)

The manuscript describes experimental measurements of Coulomb drag between MoS2 and monolayer graphene separated by few-nm thick hBN, primarily conducted by driving a current in the graphene and measuring the induced voltage in the MoS2. The authors investigate the dependence of the drag signal on temperature, carrier density and magnetic field and find substantial similarities to the available theoretical predictions. In an applied perpendicular magnetic field, both the graphene layer and the drag resistance show Shubnikov-de Haas oscillations. The magnitude of the oscillations rapidly decreases in the graphene with increasing temperature, while remaining relatively constant in the drag signal, allowing the drag resistance to act as a sensor for otherwise-inconspicuous electronic transport behavior in the graphene.

The authors effectively introduce the ideas behind the measurement and the potential significance of their results, without making overstated claims. These experiments are a nice addition to the existing body of work on Coulomb drag, and contribute new information that will be of interest to the field. The fabrication methods are consistent with the current standard for van der Waals heterostructure devices and are well-explained, and they clearly demonstrate their efforts to avoid spurious drag signals in the measurement by using a voltage-balancing bridge. There is sufficient information given in the text and supplemental material to replicate the experiment.

While the authors show Onsager reciprocity (driving one layer and measuring the drag voltage across the other, switching the drive and drag layers) in magnetotransport, it would strengthen the paper to show Onsager reciprocity in other regimes, particularly outside of the Fermi liquid drag regime. It would also be beneficial to show data from additional devices; there are 9 devices described as having been measured, but data from only 2 or 3 of them is shown. The reproducibility of the results would be more convincing with some more extensive characterization of multiple devices.

A few minor details: There is a typographical error in the legend of Figure 1h, which refers to "PVR" instead of "PADR." Also, the text refers to red arrows in Figure 1g-h that do not appear to be present.

Overall, I think this manuscript is suited for publication with minor changes.

Version 2:

Reviewer comments:

Reviewer #1

(Remarks to the Author)

The authors have adequately addressed my comments. However, the unique role of MoS₂ in the observation of pronounced SdH oscillations in the drag signal is not sufficiently elaborated. It is recommended to strengthen the relevant discussion. For example, if MoS₂ layer is replaced with graphene or other 2D materials, can similar phenomena be observed?

Reviewer #2

(Remarks to the Author)

I appreciate the authors' substantial efforts to address my previous concerns, ranging from material characterization to in-depth analysis. While a few minor points remain, they do not affect the overall quality or validity of the work. I therefore agree that the manuscript can be accepted for publication in its present form.

Reviewer #3

(Remarks to the Author)

The authors have made substantial revisions to their manuscript to address the comments and concerns of the reviewers. In this paper, the authors describe experimental signatures of Coulomb drag between monolayer graphene and bilayer MoS₂ separated by a few-nanometer-thick flake of hexagonal boron nitride. They argue that the MoS₂ acts as a sensor of the electronic compressibility of graphene, demonstrated by the drag signal exhibiting oscillations that track the Shubnikov de Haas oscillations of the graphene layer in a finite perpendicular magnetic field. The experimental results are interesting and generally support the claims made in the paper.

In the revised manuscript, the authors have satisfactorily addressed my specific comments regarding typos and the importance of showing data from additional devices. They have also provided significantly more characterization of the transport characteristics of individual layers and some proof of Onsager reciprocity. It would be more convincing to also show a full map of drag resistance as a function of bottom and top gate voltages for both graphene and MoS₂ as the drive layers, rather than just three line cuts at constant bottom gate voltage shown in Figure R4/S16 and the separated curves of drag resistance versus magnetic field in Figure R21/S20. Similar data are already shown in Figure S21 and are useful in seeing the correspondence between the two measurement configurations. Furthermore, there are clear differences between the graphene-drive and MoS₂-drive curves in Figure R21(c/d) at several temperatures. The curves are qualitatively similar, but this discrepancy should be addressed.

Overall, I think there are still a few changes that should be made, but the scientific work seems solid and the manuscript is suitable for publication with some modifications.

Author Responses to Initial Comments:

Reply to reviewer #1

General Comment:

In the manuscript, the author conducted systematic drag experiments in a graphene/MoS₂ double layer. The authors observed clear SdH-like oscillations in the drag signal that precisely track the SdH within the graphene layer. As noted by the authors, the MoS₂ layer's nearly flat compressibility facilitates the visualization of graphene's Landau quantization. While the experimental study is systematic with clear data presentation and highly readable writing, the manuscript lacks a compelling focus. The central claim in the abstract section, i.e. the observation of pronounced SdH oscillations in the drag signal, is only introduced in Figure 4. Figures 1 to 3 appear more as a simple presentation of the drag characteristics in the massless-massive system, without sufficient connection to the main finding. The authors should refine the manuscript to better highlight the key points presented in Figure 4 and strengthen the discussions on the underlying mechanisms and potential impact, which will undoubtedly improve the chances of this work being accepted.

Response:

We thank the reviewer a lot for the positive assessment of our work and for the suggestions that have helped to improve our manuscript. We have reorganized the manuscript to further bring out the main findings of this work by adding more figures and discussions on the pronounced SdH oscillations in the drag signal. Below are the point-by-point responses to the reviewer's comments.

Below are some minor comments.

Comment 1. Could the authors provide more detailed discussions on the effect of bottom gate geometry on the measured drag behaviors (Supplementary Figure 4)?

Response:

We thank the reviewer a lot for her/his suggestion. Basically, we use the local bottom gate to decrease the contact resistance of 2D semiconductor MoS₂ for all the three geometries, as shown in Figure R1. However, there are some small differences between them. In general, we can use silicon local gate to decrease the contact resistance of

MoS₂ and another bottom gold or graphite gate to tune the carrier density of monolayer graphene independently, as schematically illustrated in Figure R1g. In this geometry, the MoS₂ channel is divided into four distinct regions due to the possible misalignment of top and bottom gates: region 1 with both top and bottom gate, region 2 with only bottom gate and graphene screening, region 3 with only bottom gate, and region 4 with only silicon-gate tuning. This complicated structure would result in spurious drag signals, which are particularly pronounced near the MoS₂ band edge (Figure R1i).

To avoid this problem, we introduce only one local bottom gate to decrease the contact resistance and tune the carrier density in graphene simultaneously, as schematically shown in Figure R1a and d. However, the area of monolayer graphene in Figure R1d is larger than that of MoS₂. This configuration will prevent tuning the contact resistance of MoS₂ efficiently due to the screening induced by graphene and result in chaotic drag signals as shown in Figure R1f. Therefore, we chose the first configuration (Figure R1a) with narrow graphene flake to obtain good Ohmic contacts to MoS₂ channel and exclude spurious drag signals.

We have replaced Supplementary Figure 4 with Figure R1 and included corresponding explanations in the revised Supplementary Information, which are highlighted in blue in Page 8 and quoted below:

“Overall, the second configuration (d) will prevent tuning the contact resistance of MoS₂ efficiently due to the screening induced by graphene and result in chaotic drag signals as shown in (f). For the third configuration (g), the contact of MoS₂ is divided into four distinct regions due to the possible misalignment of top and bottom gates, which will lead to spurious drag signals presented in (i). Based on the evident comparison of the measured drag signals, we select the first device configuration for further measurements and analysis.”

Figure R1. (a) The schematic of a typical MoS₂-graphene drag device (used in the manuscript). This device configuration consists of a large local Au (or graphite) back gate and top gate. Bi/Au electrodes are used for Ohmic contact for MoS₂ and Ti/Au electrodes are used for one dimensional Ohmic contact for graphene. (b) and (c) show the corresponding zoomed-in schematic top view of the Bi/Au electrode and measured drag signals as a function of V_{bg} and V_{tg} , respectively. (d) The schematic illustrates a drag device with the graphene layer larger in size than the MoS₂ layer. The dashed line represents graphene. (e) and (f) are the corresponding zoomed-in schematic top view of the Bi/Au electrode and measured drag signals as a function of V_{bg} and V_{tg} , respectively. (g) The schematic depicts a drag device utilizing a silicon gate for tuning the carrier density in the contact region of MoS₂, while the overlapped area represents the top and bottom gates. (h) and (i) show the corresponding zoomed-in schematic top view of the Bi/Au electrode and measured drag signals as a function of V_{bg} and V_{tg} , respectively. Overall, the second configuration (d) will prevent tuning the contact resistance of MoS₂ efficiently due to the screening induced by graphene and result in chaotic drag signals as shown in (f). For the third configuration (g), the contact of MoS₂

is divided into four distinct regions due to the possible misalignment of top and bottom gates, which will lead to spurious drag signals presented in (i). Based on the evident comparison of the measured drag signals, we select the first device configuration for further measurements and analysis.

Comment 2. Have the authors checked the linear response of the drag signal to I_{drive} at low temperatures?

Response:

We thank the reviewer for the question. We have checked the response of drag signal to I_{drive} as a function of V_{tg} at $V_{\text{bg}} = 5 \text{ V}$ and $T = 77 \text{ K}$, as shown in Figure R2. From Figure R2, we can clearly see that the linear relationship between V_{drag} and I_{drive} still holds at low temperatures for different V_{tg} .

Figure R2. (a) The drag response as a function of V_{tg} for various applied drive current at $V_{\text{bg}} = 5 \text{ V}$ and $T = 77 \text{ K}$. (b) The relationship between drag response and drive current for different V_{tg} . The colored circles represent the drag voltage data extracted from (a), and the corresponding colored lines show the linear fittings for each represent V_{tg} .

In this revision, we have added Fig. R2 in the revised Supplementary Information as Supplementary Figure 15, highlighted in blue in Page 20.

Comment 3. The remarkably high passive-to-active drag ratio is intriguing. Could the authors offer a possible explanation for this observation?

Response:

We thank the reviewer a lot for the question. Typically, the passive-to-active drag ratio (PADR) is small due to the strong screening of Coulomb interactions by the electron gases. However, if the electron-electron correlation is strong or the screening of Coulomb interaction is weak in the drag system, the drag ratio would be large. For example, in some strong correlated systems such as 2D excitonic insulator (Science, 388, 274-278, 2025; Science, 388, 278-283, 2025) and double graphene or quantum wells under perpendicular magnetic field (Nature, 637, 327-332, 2025; Nature Physics, 13, 746-750, 2017; Nature, 488, 481-484, 2012), a large PADR or even "perfect" drag (approaching 1) can occur due to the formation of electron-hole pairs. But this is quite different from our case because there are no excitons in our systems.

One possible reason for the observed PADR is strong interlayer correlation induced by 2D semiconducting MoS₂. The Wigner–Seitz radius r_s correspond to the ratio between the Coulomb interactions to the kinetic energy, which can describe well the electronic correlation strength in the system. Since $r_s = m^*e^2/(4\pi\epsilon_0\epsilon\hbar^2\sqrt{\pi n_e})$ with ϵ , n_e and m^* denoting the dielectric constant, electron density and effective electron mass, large m^* , low n_e and small ϵ could be used to achieve the large r_s values (Nature 595, 53-57, 2021). For instance, the effective electron mass m^* in 2D MoS₂ is 0.4 - 0.6 (Phys. Rev. Lett. 121, 247701, 2018; Phys. Rev. B 110, L161404, 2024), which is about one order larger than that in semiconducting GaAs, and much larger than that in graphene. Thus, $r_s > 10$ can be easily obtained for an electron density of $n_e \sim 10^{11} \text{ cm}^{-2}$ in MoS₂, giving rise to stronger electronic correlation and more efficient momentum transfer in our hybrid 2D semiconductor/semimetal drag structure than conventional double quantum wells and (semi)metal/(semi)metal systems.

Another possible reason for this large PADR is the weak Coulomb screening in 2D MoS₂. In general, Thomas–Fermi wavevector q_{TF} is used to characterize how efficiency the material screens. For two-dimensional electron gas, the simple Thomas-Fermi theory leads to the long-wavelength Thomas-Fermi screening wavevector (see Reviews of modern physics, 83, 407-470, 2011),

$$q_{TF}^{2D} = \frac{2\pi e^2 D(E_F)}{\epsilon_0 \epsilon_r}.$$

The density states of MoS₂ at the Fermi level is,

$$D(E_F) = \frac{g_s g_v m^*}{2\pi \hbar^2}.$$

Therefore, the Thomas–Fermi wavevector q_{TF} of MoS₂ is,

$$q_{TF}^{MoS_2} = \frac{2m^*e^2}{\epsilon_0\epsilon_r\hbar^2},$$

where g_s and g_v are spin and valley degeneracy, and ϵ_r is background dielectric constant.

But for the monolayer graphene, the density states at the Fermi level is,

$$D(E_F) = \frac{2\sqrt{n}}{\sqrt{\pi}\hbar v_F},$$

Thus, the Thomas–Fermi wavevector q_{TF} of monolayer graphene is,

$$q_{TF}^{MG} = \frac{4\sqrt{\pi n}e^2}{\epsilon_0\epsilon_r\hbar v_F}.$$

From these equations, we can see that q_{TF} of graphene increases quickly with n and becomes much larger than the nearly constant q_{TF} of MoS₂, demonstrating the Coulomb screening in 2D MoS₂ is much weaker than that in monolayer graphene. The weak Coulomb screening in 2D MoS₂ will also give rise to the more efficient momentum transfer and larger PADR in our hybrid structure when compared with other conventional systems. (Advanced Electronic Materials, 9.3: 2201105,2023)

We have added these points in the revised main text, highlighted in blue in Page 5, and quoted below:

“This large PADR likely originates from more efficient momentum transfer, which is induced by strong interlayer correlation coupled with weak Coulomb screening within the 2D semiconductor MoS₂^{19,36,37}.”

Comment 4. The red arrows referenced in Fig. 1g-h are missing from the actual figures.

Response:

We thank the reviewer for pointing this out. We have added the red arrows in the revised Fig. 2c-d (Fig. 1g-h for old version) in the main text.

In the end, we sincerely appreciate the very helpful comments given by Referee #1. The revisions implemented based on the reviewer's suggestions have substantially improved the quality of our manuscript. We are deeply grateful for her/his support in the acceptance of this work by Nature Communications.

Reply to reviewer #2

General Comment:

Liu et al. report compressibility spectroscopy in two-dimensional systems using an asymmetric Coulomb drag configuration, particularly in the Fermi-liquid regime. The presented results are intriguing and could serve as a powerful methodology for probing subtle quantum phenomena in graphene. However, the fundamental characteristics of the active and passive layers are not clearly defined, and the terminology employed throughout the manuscript lacks clarity and consistency. The following issues should be carefully addressed before the manuscript can be considered for further consideration.

Response:

We greatly appreciate the reviewer for the positive feedback on the novelty and significance of our study. We have added some fundamental characteristics of both graphene and MoS₂ layer and refined our manuscript to improve the clarity and consistency of our findings. Below are the point-by-point responses to the reviewer's comments.

Comment 1: The completed hBN dual-gated device structure should be characterized using TEM to confirm the thicknesses of each layer (Al₂O₃, top and bottom hBN, hBN spacer, graphene, and MoS₂) and to assess the interface quality. In addition, a cross-sectional device image would greatly improve understanding, including explicit labeling of which layer acts as the active (drive) and passive (drag) layers in all figures.

Response:

We thank the reviewer a lot for her/his suggestion. Figure R3 presents the bright-field scanning transmission electron microscopy (STEM) image and the corresponding electron energy loss spectroscopy (EELS) mapping, clearly revealing the cross-sectional structure of the typical drag device with bilayer MoS₂ as the active (or passive) layer and monolayer graphene as the passive (or active) layer.

Figure R3. (a) A bright-field scanning transmission electron microscopy (STEM) image and (b) EELS mappings of the cross section of a typical graphene-MoS₂ drag device. The inset in (a) shows the high resolution high-angle annular dark-field STEM (HAADF-STEM) image of the region where bilayer MoS₂ and monolayer graphene are separated by a thin h-BN layer. All of the scale bars are 20 nm.

In this revision, we have added this image in Fig. 1 in the main text for better display of our drag device configuration.

Comment 2: Please clearly indicate whether bilayer MoS₂ is solely used and consistently serves as the passive layer throughout the manuscript. The current description is confusing.

Response:

We thank the reviewer a lot for this comment. The drag response in our system is reciprocal regardless of which layer (MoS₂ or graphene) is donated as the drive or passive layer, as shown in Fig. R4. To prevent any confusion, we have now clearly labeled the drive and passive layers in each relevant figure of the revised manuscript.

Figure R4. (a) The schematic of Onsager relation with the active (passive) layer alternated in the drag system. (b) The drag resistance as a function of V_{tg} at different V_{bg} , measured at $T = 200 \text{ K}$. The red and blue circles represent serving MoS₂ and graphene as the active layer, respectively.

We have added Figure R4 as Supplementary Figure 16 to explicitly demonstrate this Onsager reciprocity relation.

Comment 3: The drain current as a function of top- and bottom-gate voltages (V_{tg} and V_{bg}) for different temperatures should be provided separately, including both transfer and output characteristics.

Response:

We thank the reviewer a lot for her/his suggestion. We have measured the output and transfer characteristics of both MoS₂ and graphene at different temperatures. As depicted in Figures R5 and R6, the linear I - V curves of MoS₂ and graphene with different gate voltages and temperatures suggest that good Ohmic contacts are achieved in all conditions.

Figure R5. (a) The color map of the drain current in MoS₂ as a function of V_{ds} and temperature at $V_{tg} = 1\text{ V}$ and $V_{bg} = 5\text{ V}$. (b) The I - V curves of MoS₂ at different temperatures extracted from (a). The I - V curves of MoS₂ with different V_{tg} at (c) $T = 100\text{ K}$, (d) $T = 50\text{ K}$ and (e) $T = 20\text{ K}$, respectively. The applied bottom voltage is 5 V . The I - V curves of MoS₂ with different V_{bg} at (f) $T = 100\text{ K}$, (g) $T = 50\text{ K}$ and (h) $T = 20\text{ K}$, respectively. The applied top voltage is 1 V .

Figure R6. (a) The I - V curves of monolayer graphene at different temperatures for $V_{\text{tg}} = 1$ V and $V_{\text{bg}} = 5$ V. The I - V curves of monolayer graphene at $T = 20$ K for (b) $V_{\text{bg}} = 5$ V and (c) $V_{\text{tg}} = 1$ V, respectively. The I - V curves of monolayer graphene at $T = 200$ K for (b) $V_{\text{bg}} = 5$ V and (c) $V_{\text{tg}} = 1$ V, respectively.

The transfer curves of both graphene and MoS₂ at different temperatures are shown in Fig. R7. The resistance of graphene decreases as the temperature decreases, exhibiting intrinsic metallic conducting characteristics. In contrast, MoS₂ displays two distinct behaviors: its resistance decreases with increasing temperature at low V_{tg} , but decreases with decreasing temperature at high V_{tg} . This behavior clearly demonstrates the metal-insulator transition characteristic of n-type semiconductor MoS₂, consistent with previous reports.

Figure R7. The transfer curves of (a) monolayer graphene and (b) MoS₂ as a function of gate voltages for different temperatures, respectively.

In the updated Supplementary Information (changes highlighted in blue), we have replaced the original Supplementary Figure 5 with Figure R7. Additionally, Figure R5 and Figure R6 have been added as the new Supplementary Figures 6 and 7.

Comment 4: The leakage current between the active and passive layers should be presented for all measurement conditions, particularly as a function of V_{tg} , V_{bg} , temperature, and magnetic field.

Response:

We appreciate this very insightful comment from the reviewer. We have measured the leakage current between the active and passive layers at different temperatures, gate voltages and magnetic field, as shown in Figure R8. We could confirm that the leakage current is negligible during our drag measurements.

Figure R8. The leakage current I_g between the active and passive layers as a function of temperatures at $V_{tg} = 1$ V and $V_{bg} = 5$ V when the magnetic field is held for 0 T (yellow line) and 12 T (blue dots), respectively. The insets show I_g as a function of V_{tg} , V_{bg} and magnetic field.

We have added Figure R8 as Supplementary Figure 14, highlighted in blue in Page 19, to verify the validity of drag measurements.

Comment 5: Both field-effect mobility and Hall mobility as a function of temperature should be included to provide a clearer understanding of charge transport.

Response:

We thank the reviewer a lot for her/his suggestion. We have plotted the field-effect mobility of graphene and MoS₂ as a function of temperature at $V_{tg} = 1$ V and $V_{bg} = 5$ V in Figure R9. The field-effect mobility is calculated from the transfer curves in Figure R7. Note that we couldn't determine the accurate geometry factor L/W from the special geometry of our drag devices. However, we have found the geometry factor is about 1 for a similar device from the previous report (UPB Sci. Bull. Ser. A, 72, 257-271, 2010). Therefore, we adjust this approximate value for calculating the mobility of devices.

Figure R9. The field-effect mobility of (a) MoS₂ and (b) graphene for various temperatures at $V_{tg} = 1$ V and $V_{bg} = 5$ V, respectively.

In order to obtain the Hall mobility of graphene and MoS₂, we firstly measured the Hall carrier densities of these two layers as a function of temperature at $V_{tg} = 1$ V and $V_{bg} = 5$ V, as shown in Figure R10 (a) and (b), respectively. For n-type semiconductor MoS₂ ($V_{tg} = 1$ V and $V_{bg} = 5$ V), carrier concentration increases significantly with temperature because thermal energy excites more electrons to cross the energy gap. After we obtain the Hall carrier density in both graphene and MoS₂ layer, we are able to calculate the corresponding Hall mobility of these two layers for various temperatures by using the formula $\sigma = ne\mu$ and plot them in Figure R10 (c) and (d), respectively.

Figure R10. The Hall carrier density as a function of temperature in (a) MoS₂ and (b) graphene at $V_{\text{tg}} = 1 \text{ V}$ and $V_{\text{bg}} = 5 \text{ V}$, respectively. The Hall mobility in (c) MoS₂ and (d) graphene at $V_{\text{tg}} = 1 \text{ V}$ and $V_{\text{bg}} = 5 \text{ V}$ for different temperatures, respectively.

Note that the Hall and field-effect mobility values exhibit a minor difference, a consequence of the distinct measurement principles employed for each technique. We have added Figures R9 and R10 as Supplementary Figures 19 and 18 in the revised version, highlighted in blue in Pages 24 and 23, respectively.

Comment 6: The Onsager reciprocity relation should be verified and discussed to confirm the presence of genuine Coulomb drag phenomena.

Response:

We thank the reviewer for the suggestion that have helped to improve our manuscript. We agree with the reviewer on the importance of this verification. We have also discussed this in Comment 2. Our measurements have confirmed that the system's response is fully consistent with the Onsager reciprocity relation. The corresponding reciprocal relation is presented in Figure R11 (same as Fig. R4) and Supplementary Figure16, which strongly supports the genuine Coulomb drag.

Figure R11. (a) The schematic of Onsager relation with the active (passive) layer alternated in the drag system. (b) The drag resistance as a function of V_{lg} at different V_{bg} , measured at $T = 200$ K. The red and blue circles represent serving MoS₂ and graphene as the active layer, respectively.

We have incorporated Figure R11 into the Supplementary Information as the new Supplementary Figure 16, highlighted in blue in Page 21.

Comment 7: The bias condition (e.g., $I = 160$ nA) should be clarified, including why this particular condition is only considered.

Response:

We thank the reviewer a lot for her/his suggestion. As displayed in Figure 1i, the drag voltage is proportional to the drive current at 200 K, demonstrating that the drag resistance is I_{drive} - independent. For other reviewer's concern, we also employed the $I_{\text{drive}} - V_{\text{drag}}$ measurements as a function of gate voltages at low temperature to confirm the linear drag resistance, as shown in Fig. R2. To improve the signal-to-noise ratio in our measurements, we simply employ a large drive current, such as 160 nA, without any other specific intentions. For the gate bias, we typically select high bottom gate voltages, such as 4 or 5 V, to minimize the contact resistance in 2D semiconductor MoS₂ (Fig. R12) and ensure our measured drag response is reliable. Finally, we could observe the drag resistance variation as a function of carrier density (or V_{tg} when the V_{bg} is fixed), from which we were able to extract the corresponding drag resistance.

In this revision, we have added some explanation for the measured conditions in the Methods part of the manuscript, highlighted in blue in Page 10, quoted below:

“Since the relationship between V_{drag} and I_{drive} is linear, we simply employed a large drive current, such as 100 - 200 nA, to improve the signal-to-noise ratio in our drag measurements without any other specific intentions.”

Comment 8: The contact resistance should always be specified, as it directly affects both the drag and active-layer transport measurements.

Response:

We thank the reviewer for the suggestion that have helped to improve our manuscript. It is true that the contact resistance in 2D semiconductor MoS₂ is crucial for the drag transport measurements. We have performed both the two-probe and four-probe measurements of the MoS₂ and graphene layer to extract the contact resistance as a function of temperature and gate voltages, as shown in Figures R11 and R12. Note that the contact resistance unit is not normalized by multiplying channel width due to the unconventional drag device geometry. The low contact resistances ($V_{\text{tg}} = 1$ V, $V_{\text{bg}} = 5$ V) at low temperatures together with linear I - V curves in Figures R5 and R6 (Comment 3) suggest that good Ohmic contacts for both MoS₂ and graphene are achieved, enabling the reliable execution of both the drag and transport measurements.

Figure R11. (a) Two-terminal resistance R_{2p} , four-terminal resistance R_{4p} and the extracted contact resistance R_c for graphene layer in sample-S21 as a function of temperature. The contact resistance R_c for graphene layer in sample-S21 as a function of V_{tg} at (b) $T = 20$ K, (c) $T = 50$ K, and (d) $T = 100$ K, respectively. The contact resistance R_c for graphene layer in sample-S21 as a function of V_{bg} at (e) $T = 20$ K, (f) $T = 50$ K, and (g) $T = 100$ K, respectively.

Figure R12. (a) Two-terminal resistance R_{2p} , four-terminal resistance R_{4p} and the extracted contact resistance R_c for MoS₂ layer in sample-S21 as a function of temperature. The contact resistance R_c for MoS₂ layer in sample-S21 as a function of V_{tg} at (b) $T = 20$ K, (c) $T = 50$ K, and (d) $T = 100$ K, respectively. The contact resistance R_c for MoS₂ layer in sample-S21 as a function of V_{bg} at (e) $T = 20$ K, (f) $T = 50$ K, and (g) $T = 100$ K, respectively.

We have added Figures R11 and R12 as Supplementary Figures 8 and 9 in the revised version, highlighted in blue in Pages 12 and 13, respectively.

Comment 9: The electrical bias conditions and temperatures differ across the figures. Such variations must be discussed comprehensively before presenting the data, to ensure consistent interpretation. Limited fundamental experimental results currently make the discussion difficult to follow.

Response:

We thank the reviewer for the suggestion that have helped to improve our manuscript a lot. To gain a better understanding of the distinct temperature-dependent behaviors of R_{xx} and $R_{drag, xx}$ observed when probing the electron compressibility in Figures 4 and 5, we remeasured the drag response as a function of temperature for various V_{tg} values at a fixed $V_{bg} = 5$ V (consistent with the conditions in Figures 4 and 5 of the manuscript), without an applied magnetic field (as plotted in Figure R13). In consistent with the observations at $V_{bg} = 4$ V, the drag resistance follows the predicted T^2 dependence of a Fermi liquid at low temperatures, as shown in Figure R13a. As temperature increases, this behavior transitions, becoming distinctly non- T^2 and ultimately approaching a linear T dependence. This crossover range is found to be widen as MoS₂ is becoming insulating by reducing V_{tg} . These various kinds of temperature-dependent behaviors are summarized in the $T - V_{tg} - R_{drag}$ phase diagram (Figure R13b). By featuring the fitting points from the data in Figure 13a, we are able to define four distinct regimes within the phase space: the T^2 region, the T region, a $T^2 - T$ crossover region, and a region where no drag is observed.

Figure R13. (a) Temperature dependence of R_{drag} (colored open symbols) at $V_{\text{bg}} = 5$ V for different V_{tg} . The black dashed lines represent the fits to the low-temperature data with a quadratic temperature dependence, while the black solid lines correspond to the fits to the high-temperature data, assuming a linear temperature dependence. (b) The blue, yellow, and purple filled areas show the $R_{\text{drag}}(T)$ responses with T -linear, T - T^2 crossover, and T^2 behavior, respectively. Boundaries (blue and purple solid circles) are obtained by fitting the lines in (a). The open square symbols are the critical temperature T_d , which is defined as $T_d = E_F/k_B k_F d$ with k_B , k_F and d being the Boltzmann constant, Fermi vector, and the interlayer distance, respectively.

For the consistency of temperature condition during the experiments, we remeasured the carrier density as a function of V_{tg} and V_{bg} at $T = 77$ K (Figure R14), which is the same as the temperature condition in Figures 4 and 5 in the main text.

We have added Fig. R13 to the new Fig. 2 in the revised main text, highlighted in blue in Page 4.

Comment 10: Both V_{tg} and V_{bg} values should be explicitly annotated in all subfigures (e.g., Fig. 1i).

Response:

We thank the reviewer a lot for the suggestion. The corresponding V_{tg} and V_{bg} values for all measurement conditions has been annotated in the revised manuscript.

Comment 11: The meaning of the y-axis in Fig. 3d should be clearly defined.

Response:

We thank the reviewer for the suggestion. The y-axis in Fig. 3d is $\log_{10}(|R_{\text{drag}}|)$. To make sure the consistent interpretation as discussed in comment 9 and 12, we measured the carrier density and the corresponding drag response at 77 K, as shown in Figure R14. Meanwhile, in order to improve the clarity of the y-axis in Figure R14d, we have converted both x-axis and y axis to a logarithmic scale.

Figure R14. (a)-(b) The carrier density n of MoS₂ and graphene in the drag system as a function of V_{tg} and V_{bg} at $T = 77 \text{ K}$. Data are obtained by using the formula $n = B/eR_H$, where e is the elemental charge and B/R_H is obtained by extracting the slope of Hall resistance at $B = 1 \text{ T}$ and 0 T . (c) The differential carrier density δn plotted by subtracting the color map (b) with (a). Notice that black dashed line indicates the scenario of matched-density between the graphene and MoS₂ layer. (d) The map of corresponding drag resistance as a function of V_{tg} and V_{bg} at $T = 77 \text{ K}$. (e) R_{drag} plotted alongside the black dashed line in (d), which shows $1/n^2$ dependence in the matched-

density drag, suggesting the validity of massive-massless Fermions drag in strong coupling regime.

Figure R14 is the modified version of the original Figure 3, which is highlighted in blue in Page 5 of the revised manuscript.

Comment 12: The carrier density measurements in Fig. 3 (at 100 K) and the Coulomb drag data in Fig. 4 (at different temperatures) must be correlated. The Hall mobility and carrier density should be presented as a function of both different temperature and bias for consistency.

Response:

We thank the reviewer a lot for the suggestion that have helped to improve our manuscript. To better understand the drag data in Fig. 4 and ensure consistency between Fig. 3 and Fig. 4. We remeasured the carrier density as a function of gate voltage bias at $T = 77$ K, as shown in Figure R14(a) and (b). The carrier density of MoS₂ and graphene for $V_{tg} = 1$ V and $V_{bg} = 5$ V at $T = 77$ K extracted from Fig. R14(a) and (b) are about 1.18×10^{12} cm⁻² and 2.74×10^{12} cm⁻², respectively, which agree well with those from Fig. R10(a) and (b). The Hall mobility of these two layers as a function of temperature and gate voltages are shown in Figures R10 and R15, respectively. The significant difference in Hall mobility - less than 10^3 cm²/V·s for MoS₂ and higher than 10^4 cm²/V·s for graphene at low temperatures (below 77 K) under $V_{tg} = 1$ V and $V_{bg} = 5$ V, combined with the effective electron mass contrast in these two layers explain why SdH oscillations are easily detected in the graphene layer, yet the electron compressibility of the MoS₂ layer cannot be probed via transport measurements with a 12 T field. Therefore, the MoS₂ layer, which shows flat electron compressibility at 12 T, is effectively employed as a sensor for probing the electron compressibility of the high-mobility graphene.

Figure R15. The Hall mobility as a function of V_{tg} in (a) MoS₂ and (b) graphene at $V_{bg} = 5$ V and $T = 77$ K.

Figure R15 has been included in the revised Supplementary Figure 18 (highlighted in blue) to improve the consistency of presentation across Figures 4 and 5 in the main text.

Comment 13: In Fig. 1b, the scale bar indicates 10 mm, please confirm if this is a typo.

Response:

We thank the reviewer a lot for pointing out this mistake. The scale bar in Fig. 1b is indeed a typo and has been corrected to 10 μ m in the revised manuscript.

Comment 14: In Fig. 1c, the data points corresponding to G_{MoS2} for $V_{tg} = -3$ V to -1 V appear missing. In addition, including a semi-logarithmic scale would help reveal the n-type semiconducting behavior of bilayer MoS₂.

Response: We thank the reviewer a lot for the comment. In general, we use an ac lock-in transport method to measure the resistance (or conductance) of the MoS₂ layer, which is different from a dc measurement. Due to the semiconducting nature of MoS₂, its resistance increases rapidly as the Fermi level approaches the band gap, as presented in Fig. R16(a) and (b). This characteristic causes the lock-in amplifier's phase to rise above 10°, and in some cases even change sign (Fig. R16c). Therefore, we regard

signals acquired under such large phase shifts ($> \pm 10^\circ$) as unreliable and treat them as inaccurate measurements of the intrinsic MoS₂ resistance.

Figure R16. (a) The field effect curves of MoS₂ measured by the ac lock-in method at $T = 200$ K with different bottom gates. (b) The conductance of MoS₂ as a function of V_{tg} in a semi-logarithmic scale at $T = 200$ K with different bottom gates. (c) The phase of lock-in amplifier for the corresponding conductance signals in (a) or (b). The colored arrows in these figures indicate the turning points beyond which the intrinsic conductance of MoS₂ can be measured accurately.

In the revision, we have added Figure R16 in original Supplementary Figure 6 as the new Supplementary Figure 10, highlighted in blue in Page 15.

Comment 15: In Fig. 1f, please define the parameter R_{MoS_2} . In Fig. 1h, define PVR explicitly. Numerous typographical errors are present throughout the manuscript and should be corrected carefully.

Response:

We thank the reviewer a lot for her/his suggestion. R_{MoS_2} is defined as the lateral resistance of MoS₂ layer. The PVR is one typo error, and should be the passive-to-active drag ratio (PADR), which is defined as $I_{\text{drag}}/I_{\text{drive}} = R_{\text{drag}}/R_{\text{passive layer}}$. We have carefully modified our manuscript to make it more accurate.

Comment 16: In Fig. S6c, the V_{tg} -dependent MoS₂ resistance behavior is nearly symmetric and resembles that of graphene, suggesting ambipolar characteristics. Please elaborate on this observation.

Response:

We thank the reviewer for the comment. As we have already discussed in Fig. R16, the “ambipolar-like” transfer curves observed in MoS₂ are mainly caused by spurious phase signals that appear when the Fermi level approaches the band edge of the two-dimensional semiconductor MoS₂ during the ac lock-in measurements (the same data is plotted in Fig. R17). Therefore, In Fig. R17(a), the accurate resistance of MoS₂ is represented solely by the signals on the right side of the resistance peak, as marked by the colored arrows.

Figure R17. (a) The field effect curves of MoS₂ measured by the ac lock-in method at $T = 200$ K with different bottom gates. (b) The phase of lock-in amplifier for the corresponding resistance signals in (a). Since the data with a phase exceeding $\pm 10^\circ$ are typically considered unreliable, the intrinsic conductance of MoS₂ can only be obtained beyond the colored arrows in (a).

We have incorporated Figure R17 in original Supplementary Figure 6 as the new Supplementary Figure 10, highlighted in blue in Page 15.

Comment 17: In Fig. S10a, G_{MoS_2} measured at $V_{\text{bg}} = 4$ V is lower than that at $V_{\text{bg}} = 1$ V (Fig. 1c), which is counterintuitive for n-type MoS₂. Please clarify this discrepancy. If different devices were used, please specify device numbers in all figures.

Response:

We appreciate the reviewer a lot for pointing this out. The y-axis tick values contain a typographical error in Supplementary Figure 10a. The number 80 should be 100, so that the conductance of MoS₂ at $V_{\text{bg}} = 4$ V is larger than that at $V_{\text{bg}} = 1$ V. The conductance of MoS₂ versus V_{tg} for different V_{bg} is plotted in Figure R18, which clearly exhibits n-type conduction.

Figure R18. The comparison of the V_{tg} dependence of G_{MoS_2} , dG/dV_{tg} and R_{drag} . (a) The conductance of MoS₂ (G_{MoS_2}), (b) the MoS₂ channel conductance derivative with respect to gate voltage (dG/dV_{tg}), and (c) the drag resistance (R_{drag}) of the device as a function of V_{tg} at $T = 200$ K and $V_{\text{bg}} = 4$ V. (d) G_{MoS_2} , (e) dG/dV_{tg} , and (f) R_{drag} as a function of V_{tg} at $T = 200$ K and $V_{\text{bg}} = 6$ V. The green regions in the figures indicate the onset of dG/dV_{tg} , within which the maximum value of R_{drag} is found.

We have already corrected this typographical error in the revised Supplementary Figure 13, highlighted in blue in Page 18.

Comment 18: The thickness of the hBN spacer is a critical parameter for interlayer coupling at the Gr/hBN/MoS₂ interface. The electron compressibility as a function of hBN thickness should be presented and discussed. In addition, please discuss the case

where the positions of MoS₂ and graphene are swapped. Why is MoS₂ always placed on top of graphene? Addressing this question would strengthen the physical reasoning.

Response:

We thank the reviewer a lot for the suggestion that have helped to improve our manuscript. We totally agree with the reviewer that the thickness of the hBN spacer d is a critical parameter for drag response. Accordingly, we have measured the dual-gate mapping of drag resistance for three typical samples with different interlayer spacing, as shown in Fig. R19(a) to (c). We could clearly see that the magnitude of drag resistance is inversely proportional to d , which is consistent with previous report (Nature Physics, 8, 896-901, 2012; Nature communications, 5, 5824, 2014). For better comparison, we plotted the drag resistance as a function of carrier density for samples S21 and S15 at 77 K in Fig. R19(d). Note that we couldn't extract the exact values of Hall carrier density for sample S24 due to the insufficient electrodes. Moreover, we utilized drag and transport measurements at 77 K to probe the electron compressibility of graphene in two drag systems with two interlayer spacing d , as presented in Fig. R19(e). The carrier densities were fixed at $1.2 \times 10^{12} \text{ cm}^{-2}$ for MoS₂ and $2.7 \times 10^{12} \text{ cm}^{-2}$ for graphene, respectively. The system with the thinner spacer (sample S21) exhibited stronger interlayer coupling, resulting in a higher magnitude of oscillations in the drag signal compared to the weaker interlayer coupling of the thicker spacer (sample S15). This significant difference indicates that the thickness of the spacer is critically important for efficiently probing compressibility in the drag system.

Figure R19. The drag resistance as a function of V_{tg} and V_{bg} for three samples, labeled (a) S21, (b) S24 and (c) S15, which feature different interlayer h-BN spacer thicknesses. (d) The drag resistance as a function of carrier density in MoS₂ for samples S21 and S15 with a same carrier density in graphene at 77 K. (e) A comparison of intrinsic resistance of graphene and drag resistance as a function of magnetic field between samples S21 and S15. The data is measured at 77 K with the same fixed carrier density in both the graphene and MoS₂ layer.

We have addressed this question and added Fig. R19 as Supplementary Figure 14. More discussion could be found in the revised manuscript, highlighted in blue in Page 9, quoted below:

“The final section addresses the dependence of the drag response on the spacer thickness d , a key factor determining the strength of interlayer coupling within the drag system. We have measured the dual-gate mapping of drag resistance for three typical samples with different interlayer spacing, as shown in Supplementary Figure 24(a) to (c). We could clearly see that the drag response is reproducible in different samples, and the magnitude of drag resistance is inversely proportional to d , which is consistent with previous report^{3, 27}. For better comparison, we plotted the drag resistance as a

function of carrier density for samples S21 and S15 at 77 K in Supplementary Figure 24(d). Moreover, we utilized drag and transport measurements at 77 K to probe the electron compressibility of graphene in two drag systems with different d , as presented in Supplementary Figure 24(e). The carrier densities for this measurement were fixed at $\sim 1.2 \times 10^{12} \text{ cm}^{-2}$ for MoS₂ and $\sim 2.7 \times 10^{12} \text{ cm}^{-2}$ for graphene, respectively. The system with the thinner spacer (sample S21) exhibited stronger interlayer coupling, resulting in a higher magnitude of oscillations in the drag signal compared to the weaker interlayer coupling of the thicker spacer (sample S15). This significant difference indicates that the thickness of the spacer is critically important for efficiently probing compressibility in the drag system.”

Now, we discuss why we always put MoS₂ on the top of graphene. In order to achieve Ohmic contact with 2D semiconductor MoS₂ and improve the electron mobility in it, we have adopted window-contact technique as previously reported. We firstly etched through the windows on top h-BN and use it to pick up MoS₂ and other layers afterwards. Then, we adopted Bismuth with low work function as electrodes contacted to MoS₂ and achieved Ohmic contact even at low carrier density and temperatures. Therefore, we have to make the hybrid drag structure with MoS₂ placed on the top of graphene, as illustrated in Supplementary Figures 1 and 3. Although the fabrication would be super challenging if the MoS₂ and graphene layers were swapped, the drag response would remain identical owing to the vertically mirror-symmetric configuration.

We have added some explanation in the revised main text and Methods part to address this question, highlighted in blue in Pages 2 and 10, respectively, quoted below:

“This requires the MoS₂ to be the top layer in the heterostructure to facilitate the fabrication process in this study.”

“Note that we always put 2D semiconductor MoS₂ on the top of graphene because we adopted window-contact technique as previously reported³¹. We firstly etched through the windows on top h-BN and use it to pick up MoS₂ and other layers afterwards. Then, we adopted Bismuth with low work function as electrodes contacted to MoS₂ and achieved Ohmic contact at low carrier density and temperatures. Although the fabrication would be more challenging if the MoS₂ and graphene layers were swapped, the drag response would remain identical due to the vertically mirror-symmetric configuration.”

Comment 19: The contact resistance for both MoS₂ and graphene as a function of bias and temperature should be presented to ensure accurate analysis of interlayer transport behavior.

Response:

We thank the reviewer for the comment. We have performed both the two-probe and four-probe measurements of the MoS₂ and graphene layer to extract the contact resistance as a function of temperature and gate voltages, as shown in Figures R11 and R12. The low contact resistances ($V_{\text{tg}} = 1 \text{ V}$, $V_{\text{bg}} = 5 \text{ V}$) at low temperatures together with linear I - V curves in Figures R5 and R6 suggest that good Ohmic contacts for both MoS₂ and graphene are achieved, enabling the reliable execution of both the drag and transport measurements.

We have added Supplementary Figures 8 and 9 in the revised Supplementary Information to discuss the contact resistance of the both layers.

Comment 20: Since the effective mass of MoS₂ is roughly an order of magnitude larger than that of graphene, it would be helpful to include an intuitive schematic illustration showing how the Shubnikov–de Haas oscillations in MoS₂ could be amplified or modulated through electron compressibility.

Response:

We thank the reviewer a lot for the suggestion. The MoS₂ layer exhibits low electron mobility due to its high electron effective mass and high concentration of defects and disorders. As schematically illustrated in Fig. R20, the low mobility establishes a diffusive transport model in MoS₂ with a magnetic field applied. Electrons move randomly between scattering disorders, characterized by non-circular, jagged paths in the top layer. This diffusive conducting model prevents the observation of significant SdH oscillations under 12 T, meaning the electron compressibility in MoS₂ is nearly flat as a function of the magnetic field.

In stark contrast, high-mobility graphene enters the quantum regime under low temperatures and high magnetic fields. Electrons primarily follow cyclotron orbits perpendicular to the magnetic field (in the bottom layer). In other words, the electron states could be easily quantized into distinct Landau Levels (LLs). Consequently, the electron compressibility in graphene oscillates with the magnetic field for a fixed carrier density, allowing for the observation of evident SdH oscillations. However, as the

temperature increases, transport across the sample occurs via inter-orbit scattering (the dashed lines connecting the orbits), which is a diffusive process, leading to the amplitude of the SdH oscillations to decrease until they vanish.

But for the case of drag system, the amplitude of magneto-drag resistance oscillations is significantly amplified by the screening interlayer interaction $|U_{12}|^2$, and the temperature dependence is extremely different from single-layer transport, allowing the oscillations to remain visible even at high temperatures. Furthermore, the flat electron compressibility in MoS₂ confirms the origin of the observed oscillations in R_{drag} is solely from the graphene layer, suggesting that MoS₂ serves as an excellent sensor of probing electron compressibility in high mobility materials.

Figure R20. The schematic illustration of diffusive conductive model in low mobility MoS₂ layer with flat electron compressibility and quantized electron states in high mobility graphene layer with oscillatory electron compressibility.

In this revision, we have incorporated Figure R20 in Figure 5, and added the corresponding description of this schematic for better understanding the asymmetric drag system, highlighted in blue in Page 9.

In summary, while the manuscript presents an interesting approach to probing compressibility in two-dimensional systems via Coulomb drag, it currently lacks structural, electrical, and conceptual consistency. The figures require systematic reorganization, and several critical parameters (bias, temperature, and layer identity) must be clarified before the study can provide a coherent physical picture. If the manuscript provides a more detailed discussion of the fundamental charge transport

characteristics in each layer, a deeper and more quantitative review of the Coulomb drag phenomena would be possible.

Response:

We are very grateful for Referee#2's positive assessment and kind comments for our manuscript. By adding a more detailed discussion of the fundamental charge transport characteristics in each layer, the revised manuscript indeed provides a deeper understanding of the novel drag phenomenon. Her/his constructive suggestions have strengthened the overall quality of our work. We sincerely appreciate the reviewer's positive evaluation regarding the novelty and significance of our work.

Reply to reviewer #3

General Comment:

The manuscript describes experimental measurements of Coulomb drag between MoS₂ and monolayer graphene separated by few-nm thick hBN, primarily conducted by driving a current in the graphene and measuring the induced voltage in the MoS₂. The authors investigate the dependence of the drag signal on temperature, carrier density and magnetic field and find substantial similarities to the available theoretical predictions. In an applied perpendicular magnetic field, both the graphene layer and the drag resistance show Shubnikov-de Haas oscillations. The magnitude of the oscillations rapidly decreases in the graphene with increasing temperature, while remaining relatively constant in the drag signal, allowing the drag resistance to act as a sensor for otherwise-inconspicuous electronic transport behavior in the graphene.

The authors effectively introduce the ideas behind the measurement and the potential significance of their results, without making overstated claims. These experiments are a nice addition to the existing body of work on Coulomb drag, and contribute new information that will be of interest to the field. The fabrication methods are consistent with the current standard for van der Waals heterostructure devices and are well-explained, and they clearly demonstrate their efforts to avoid spurious drag signals in the measurement by using a voltage-balancing bridge. There is sufficient information given in the text and supplemental material to replicate the experiment.

Response:

We thank the reviewer a lot for her/his positive assessment of our work. Below, we provide detailed responses to each of the reviewer's comments in a point-by-point manner.

Comment 1: While the authors show Onsager reciprocity (driving one layer and measuring the drag voltage across the other, switching the drive and drag layers) in magnetotransport, it would strengthen the paper to show Onsager reciprocity in other regimes, particularly outside of the Fermi liquid drag regime.

Response:

We thank the reviewer a lot for the suggestion. We present here additional data comparing the drag response by switching the drive and drag layers. Fig. R21(a) and (b) show the color plots of magneto-drag resistance versus magnetic field at different gate voltages and $T = 200$ K, using the MoS₂ as the drive layer and graphene as the drag layer respectively. The Onsager reciprocity has been confirmed valid even though the drag transport is in the non-Fermi liquid regime. In addition, we have measured the magneto-drag response for reciprocal drag circuit configurations at the same gate voltages, as presented in Fig. R21(c) and (d). The general degree to which the two data sets coincide measurements provide strong evidence that Onsager reciprocity is valid for our devices in either Fermi liquid or non-Fermi liquid regime.

Figure R21. The drag resistance versus magnetic field for varying V_{tg} when (a) MoS₂ and (b) graphene serving as the drive layer, respectively. The drag resistance as a function of magnetic field at the same gate voltage ($V_{\text{bg}} = 4.1$ V and $V_{\text{tg}} = 1$ V) but different temperatures, for reciprocal layer configurations: (c) MoS₂ drag graphene and (d) graphene drag MoS₂. The curves in (c) and (d) is vertically shifted for better visualization.

We have added Figure R21 as Supplementary Figure 20 in the revised Supplementary Information, highlighted in blue in Page 25.

Comment 2: It would also be beneficial to show data from additional devices; there are 9 devices described as having been measured, but data from only 2 or 3 of them is shown. The reproducibility of the results would be more convincing with some more extensive characterization of multiple devices.

Response:

We thank the reviewer a lot for her/his valuable suggestion. Note that not all these devices survive the process or retain sufficient functional electrodes for subsequent measurements. The reproducible drag data of new samples could be found in the Fig. R19. As for the results of probing electron compressibility by drag measurements, Fig. R22 shows the longitudinal channel resistance of graphene and the magneto-drag resistance data from another typical device sample S24 for an additional comparison. The geometry of sample S24 is the same as sample S21. In order to obtain good Ohmic contact of MoS₂ in the devices, we applied a large V_{bg} (7.5 V), and measure the R_{xx} of graphene as a function of magnetic field for different temperatures. One could observe evident oscillations of R_{xx} in graphene at low temperatures in Fig. R22(a). Apparently, the amplitude of these Shubnikov–de Haas oscillations diminishes as the temperature rises, leading to a smooth resistance background when the temperatures exceed 100 K. For comparison, we have plotted the $R_{drag, xx}$ for reciprocal drag circuit configurations under the same measurement conditions as the graphene channel, as shown in Fig. R22(b) and (c). Notably, the oscillation amplitudes in the drag response do not vanish even as the temperature increases to 140 K, confirming our main finding: the reproducible persistence of SdH-like oscillations in the drag signal even when graphene’s own transport is featureless. The results are fully consistent with those of sample S21 in the main manuscript.

Figure R22. (a) R_{xx} of graphene as a function of magnetic field for different temperatures. $R_{\text{drag}, xx}$ as a function of magnetic field for different temperatures, where (b) MoS₂ and (c) graphene serve as the drive layer. All data are taken at $V_{\text{bg}} = 7.5$ V and $V_{\text{tg}} = 1$ V.

In this revision, we have added Figures R19 and R22 as Supplementary Figure 24 and 23, respectively, in the updated Supplementary Information. These figures present the reproducible results obtained from samples other than Sample S21, which is primarily discussed in the main text.

Comment 3: A few minor details: There is a typographical error in the legend of Figure 1h, which refers to “PVR” instead of “PADR.” Also, the text refers to red arrows in Figure 1g-h that do not appear to be present.

Response:

We appreciate the reviewer for pointing these out. We have corrected these typographical errors and modified the figures in the revised manuscript.

Overall, I think this manuscript is suited for publication with minor changes.

Response:

We sincerely thank Referee#3 for the thorough and insightful comments. The revisions according to the reviewer’s suggestions have improved the quality of our manuscript a lot. We appreciate her/his time in evaluating our study and support of publication in Nature Communications.

Author Responses to Initial Comments:

Reply to reviewer #1

Comment:

The authors have adequately addressed my comments. However, the unique role of MoS₂ in the observation of pronounced SdH oscillations in the drag signal is not sufficiently elaborated. It is recommended to strengthen the relevant discussion. For example, if MoS₂ layer is replaced with graphene or other 2D materials, can similar phenomena be observed?

Response:

We thank the reviewer a lot for the comment that have helped to improve our manuscript. It is a great question that whether similar phenomena could be observed when the MoS₂ is replaced by other 2D materials.

We have expanded relevant discussions on the role of the MoS₂ layer and evaluated the consequence of replacing MoS₂ with other 2D materials in the Page 9 of the main text, highlighted in blue, quoted below:

“The unique role of MoS₂ in the observation of pronounced SdH oscillations is of two origins, i.e., relatively flat electron compressibility in its own, and sufficiently strong Coulomb interaction factor $|U_{12}|$. In this regard, if MoS₂ in the current system is replaced by other 2D semiconductor with flat electron compressibility - such as MoSe₂ and WSe₂, provided that Ohmic contacts could be obtained at low temperatures - should theoretically exhibit similar phenomena. Otherwise, for instance, replacing MoS₂ with graphene, high screening in both layers will lead to a dramatic reduction of interlayer Coulomb interaction, and their SdH signals tend to be entangled, complicating the decoupling of layer-specific information.”

While that SdH oscillations in the drag signal more evident than intrinsic transport at elevated temperatures.

Reply to reviewer #2

General Comment:

I appreciate the authors' substantial efforts to address my previous concerns, ranging from material characterization to in-depth analysis. While a few minor points remain, they do not affect the overall quality or validity of the work. I therefore agree that the manuscript can be accepted for publication in its present form.

Response:

We are deeply grateful for her/his support in the acceptance of this work by Nature Communications.

Reply to reviewer #3

General Comment:

The authors have made substantial revisions to their manuscript to address the comments and concerns of the reviewers. In this paper, the authors describe experimental signatures of Coulomb drag between monolayer graphene and bilayer MoS₂ separated by a few-nanometer-thick flake of hexagonal boron nitride. They argue that the MoS₂ acts as a sensor of the electronic compressibility of graphene, demonstrated by the drag signal exhibiting oscillations that track the Shubnikov de Haas oscillations of the graphene layer in a finite perpendicular magnetic field. The experimental results are interesting and generally support the claims made in the paper.

Response:

We greatly appreciate the reviewer for the positive assessment of our study. Below are the point-by-point responses to the reviewer's comments.

Comment 1:

In the revised manuscript, the authors have satisfactorily addressed my specific comments regarding typos and the importance of showing data from additional devices. They have also provided significantly more characterization of the transport characteristics of individual layers and some proof of Onsager reciprocity. It would be more convincing to also show a full map of drag resistance as a function of bottom and top gate voltages for both graphene and MoS₂ as the drive layers, rather than just three linecuts at constant bottom gate voltage shown in Figure R4/S16 and the separated curves of drag resistance versus magnetic field in Figure R21/S20. Similar data are already shown in Figure S21 and are useful in seeing the correspondence between the two measurement configurations.

Response:

We thank the reviewer a lot for her/his suggestion. Here we have included the full maps of drag resistance of sample-S21 as a function of bottom and top gate voltages as shown in Figure R1, with graphene and MoS₂ serving as the drive layer, respectively.

Figure R1. Dual-gate maps of the drag resistance of sample-S21 at $T = 200$ K with (a) the graphene layer and (b) the MoS₂ layer serving as the active layer, respectively. Certain regions in the maps (a) and (b) are masked because they fell out of phase with the lock-in amplifier, as seen in Supplementary Figure 12. (c) The schematic of Onsager relation with the active (passive) layer alternated in the drag system. (d) The drag resistance as a function of V_{tg} at different V_{bg} , measured at $T = 200$ K. The red and blue circles represent serving MoS₂ and graphene as the active layer, respectively.

In addition, as shown in Figure R2 (a) and (b), we have added the full maps of drag resistance as a function of bottom gate and magnetic field with graphene and MoS₂ serving as the drive layer, respectively. The general high degree of correspondence

between the R_{drag} maps for different drive layers (Figures R1 and R2) provides strong evidence that Onsager reciprocity is respected in this regime.

Figure R2. Dual-gate maps of the drag resistance of sample-S24 at $T = 200$ K with (a) the graphene layer and (b) the MoS₂ layer serving as the drive layer. The drag resistance versus magnetic field for varying V_{bg} when (c) graphene and (d) MoS₂ serving as the drive layer, respectively. The drag resistance as a function of magnetic field at the same gate voltage ($V_{bg} = 4.1$ V and $V_{tg} = 1$ V) but different low temperatures, for reciprocal layer configurations: (e) graphene drag MoS₂ and (f) MoS₂ drag graphene. The curves in (e) and (f) is vertically shifted for better visualization. Note that the curves in (e) and (f) for reciprocal configurations do not overlap perfectly; this deviation may be induced by disorder within the device.

For better showing the Onsager reciprocal relation in our drag systems, we have added the full maps of drag response for reciprocal configurations in Supplementary Figures 16 and 20, highlighted in blue in Pages 21 and 25.

Comment 2:

Furthermore, there are clear differences between the graphene-drive and MoS₂-drive curves in Figure R21(c/d) at several temperatures. The curves are qualitatively similar, but this discrepancy should be addressed.

Response:

We thank the reviewer a lot for her/his comment that have helped to improve our manuscript. The reviewer correctly notes that while the trends in Figure R21 (c) and (d) are highly similar, the curves do not overlap perfectly. While the precise origin of this discrepancy remains under investigation, the deviation from ideal reciprocity may stem from disorder within specific devices, which can contribute a small rectified component to the drag signal under low temperatures and magnetic fields.

We have addressed the discrepancy regarding the reciprocal relation in the main text (Page 8) and the caption of Figure S20 in the revised Supplementary Information (Page 26), highlighted in blue.

“The Onsager reciprocity relation for magnetodrag resistance remains nearly valid regardless of whether the system is within or outside the Fermi liquid regime (as shown in Supplementary Figure 20 and 21). It is noteworthy that a slight departure from perfect reciprocity occurs at low temperatures and under magnetic fields. While the microscopic origins of this non-ideal reciprocal Coulomb drag remain unclear, they

may stem from a high concentration of disorder in specific devices, which can introduce a small rectified component to the drag signal.”

“Note that the curves in (e) and (f) for reciprocal configurations do not overlap perfectly; this deviation may be induced by disorder within the device.”

Overall, I think there are still a few changes that should be made, but the scientific work seems solid and the manuscript is suitable for publication with some modifications.

Response:

We are very grateful for the reviewer’s positive assessment and kind comments for our manuscript. Her/his constructive suggestions have strengthened the overall quality of our work. We sincerely appreciate the reviewer’s positive evaluation regarding the novelty and significance of our work.